# Diagnostic and Therapeutic Approaches for Glioblastoma and Neuroblastoma Cancers Using Chlorotoxin Nanoparticles

**DOI:** 10.3390/cancers15133388

**Published:** 2023-06-28

**Authors:** Taahirah Boltman, Mervin Meyer, Okobi Ekpo

**Affiliations:** 1Department of Medical Biosciences, University of the Western Cape, Robert Sobukwe Road, Bellville, Cape Town 7535, South Africa; 2Department of Science and Innovation/Mintek Nanotechnology Innovation Centre, Biolabels Node, Department of Biotechnology, University of the Western Cape, Robert Sobukwe Road, Bellville, Cape Town 7535, South Africa; memeyer@uwc.ac.za; 3Department of Anatomy and Cellular Biology, College of Medicine and Health Sciences, Khalifa University, Abu Dhabi P.O. Box 127788, United Arab Emirates

**Keywords:** blood-brain-barrier (BBB), chlorotoxin (CTX), glioblastoma (GB), matrix metalloproteinase 2 (MMP-2), nanoparticles (NPs), neuroblastoma (NB)

## Abstract

**Simple Summary:**

Glioblastoma multiforme (GB) and neuroblastomas (NBs) are nervous system cancers that are difficult to diagnosis and treat. Chlorotoxin (CTX), is a peptide extracted from scorpion venom which easily binds with many cancer cells, especially in GB and NB. Through nanotechnological methods, CTX can be conjugated to nanoparticles (NPs), and used both as for both diagnostic and therapeutic (theranostics) applications. This review article discusses the potential use of CTX-NP formulations for GB and NB, provides the current understanding of the mechanisms by which CTX may cross the difficult blood-brain barrier to target tumour cells. The authors extensively discuss the current state of research involving similar formulations and suggest areas for further investigation, such as using CTX-NPs for hyperthermia-based treatments therapy. Furthermore, the article discusses future trends and perspectives for novel CTX-based NP formulations to revolutionize the diagnosis and treatment of these challenging brain tumours.

**Abstract:**

Glioblastoma multiforme (GB) and high-risk neuroblastoma (NB) are known to have poor therapeutic outcomes. As for most cancers, chemotherapy and radiotherapy are the current mainstay treatments for GB and NB. However, the known limitations of systemic toxicity, drug resistance, poor targeted delivery, and inability to access the blood-brain barrier (BBB), make these treatments less satisfactory. Other treatment options have been investigated in many studies in the literature, especially nutraceutical and naturopathic products, most of which have also been reported to be poorly effective against these cancer types. This necessitates the development of treatment strategies with the potential to cross the BBB and specifically target cancer cells. Compounds that target the endopeptidase, matrix metalloproteinase 2 (MMP-2), have been reported to offer therapeutic insights for GB and NB since MMP-2 is known to be over-expressed in these cancers and plays significant roles in such physiological processes as angiogenesis, metastasis, and cellular invasion. Chlorotoxin (CTX) is a promising 36-amino acid peptide isolated from the venom of the deathstalker scorpion, *Leiurus quinquestriatus*, demonstrating high selectivity and binding affinity to a broad-spectrum of cancers, especially GB and NB through specific molecular targets, including MMP-2. The favorable characteristics of nanoparticles (NPs) such as their small sizes, large surface area for active targeting, BBB permeability, etc. make CTX-functionalized NPs (CTX-NPs) promising diagnostic and therapeutic applications for addressing the many challenges associated with these cancers. CTX-NPs may function by improving diffusion through the BBB, enabling increased localization of chemotherapeutic and genotherapeutic drugs to diseased cells specifically, enhancing imaging modalities such as magnetic resonance imaging (MRI), single-photon emission computed tomography (SPECT), optical imaging techniques, image-guided surgery, as well as improving the sensitization of radio-resistant cells to radiotherapy treatment. This review discusses the characteristics of GB and NB cancers, related treatment challenges as well as the potential of CTX and its functionalized NP formulations as targeting systems for diagnostic, therapeutic, and theranostic purposes. It also provides insights into the potential mechanisms through which CTX crosses the BBB to bind cancer cells and provides suggestions for the development and application of novel CTX-based formulations for the diagnosis and treatment of GB and NB in the future.

## 1. Introduction

Cancer is the leading cause of low life expectancy and death worldwide [1]. Nervous system (NS) tumors include a broad spectrum of brain and spinal cord malignancies that contribute to global economic burden and are often associated with short- and long-term disabilities [2]. The treatment of malignant tumors of the NS remains a challenge due to the BBB, with about 330,000 new central nervous system (CNS) cancer cases reported globally [3]. High-grade glioblastoma, also known as glioblastoma multiforme (GB), is the most aggressive and consistently debilitating primary brain tumor in adults, with a dismal median survival time of 12–15 months and a 5-year survival rate of less than 7% [4]. Standard treatments are ineffective due to the diffuse invasion and infiltrative overgrowth of heterogeneous glioma cells contributing to the development of irregular and indistinct tumor margins, thereby hindering complete surgical resection [5]. The location of deep-seated GB also makes it difficult to treat without damaging healthy brain cells, hence radiotherapy and chemotherapeutic drugs have the treatment limitations of non-specificity and systemic toxicity [6], in addition to the most important challenge of BBB permeability, i.e., effective delivery into the brain. [7].

Neuroblastoma (NB) are amongst the most frequent childhood cancers of the sympathetic NS and remains one of the major challenges in pediatric oncology [8]. The progression of NB is associated with hematogenous metastasis, common relapses, and a fast decline in survival timeline and drug resistance [9]. As for GB, conventional treatment for NB is limited by systemic toxicity and more than half of children diagnosed with high-risk NB do not respond to high-dose chemotherapy and often demonstrate multi-drug resistance [10]. Some of the reported adverse side effects of high-dose chemotherapy include nephrotoxicity, cardiotoxicity, and gonadotoxicity leading to infertility in later life [11,12]. Specific long-term toxic side effects include cognitive deficits, epilepsy, growth reduction, thyroid function disorders, learning difficulties, and an increased risk of secondary cancers in survivors of high-risk NB [13,14,15,16]. Although there have been improvements in the management of NB during the past two decades, the overall cure rate remains at approximately 50% for high-risk patients [10]. Tumor migration and invasion have been identified as the major causes of treatment failure in patients with malignant tumors [17].

Metastasis is a multistep process that consists of cancer cell migration and invasion [18] and most NS tumors are known to undergo metastasis similar to other cancer types [19]. Tumor metastasis comprises neovascularization as well as cell adhesion, invasion, migration, and proliferation, with the degradation of the extracellular matrix (ECM) and the basement membrane playing an important role [20,21]. The overexpression of matrix metalloproteinases (MMPs) in tumor cells has been implicated in tumor progression [22]. Metalloproteinases (MPs) are a family of secreted, zinc-dependent endopeptidases involved in such processes as tissue-remodelling, wound healing, embryo implantation, tumor invasion, metastasis, and angiogenesis [23,24,25]. MMPs are reported to contribute to the degradation of extracellular matrix (ECM), stromal connective tissue, and the BBB tight junctions, which are some of the driving factors of cancer invasion and metastasis, the progression of neurodegenerative diseases, and other pathological disorders [26].

There are over 20 matrix metalloproteinase (MMP) family members, and the subset of MMP-2s, released from neurons and neuroglia, is known to be present in the CNS [27,28]. Literature evidence suggests that the over-expression of MMP-2 is an active contributor to the progression of malignant GB [29,30,31] and NB [32,33,34] via increased cancer-cell growth, migration, invasion, and angiogenesis. The degree of invasiveness of GB and NB tumors is related to increased levels of MMP-2 expression [33,35]. In addition to the overexpression of MMP-2 on the cell surface, the chloride ion channel 3 (ClC-3) is specifically upregulated in human GB cells but not in normal glial cells and neurons [36,37]. ClC-3 is involved in cell cytoskeletal rearrangements as well as cell shape and movements during cell migration [36,38].

The annexin protein family is a group of calcium-dependent phospholipid-binding proteins that contribute to such cellular functions as angiogenesis, apoptosis, cell migration, proliferation, invasion, and cohesion [39,40]. The surface protein annexin A2, a calcium-binding cytoskeletal protein located at the extracellular surface of various tumor cell types, including glioma, is involved in tumor progression through cell migration and invasion [41,42]. Annexin A2 is also implicated in enhanced multidrug resistance in NB [43]. 

Thus, MMP-2, chloride channels, and Annexin 2 are all involved in malignant cell migration and invasion and provide therapeutic opportunities for targeting GB and NB cancers. One promising molecule for achieving such targeting is the peptide, chlorotoxin (CTX) which is a 36-amino acid peptide isolated from the venom of the deathstalker scorpion, *Leiurus quinquestriatus*, known to specifically bind to gliomas and many other tumors of neuroectodermal origin similar to NB [44]. Although the precise mechanism of CTX targeting has yet to be fully elucidated, a number of studies have suggested the presence of many targeting receptors for CTX on the surfaces of different cancer cells, including chloride channels [45,46,47], MMP-2 [48,49,50], annexin A2 [51,52], estrogen receptor alpha (ERα) [53] and neuropilin-1 receptor (NRP-1) [54,55]. In gliomas, it is reported that MMP-2 and CIC-3 form a protein complex located within the same membrane domain targeted by CTX and could potentially inhibit glioma cell invasion through the induction of MMP-2/ClC-3 protein complex endocytosis [48]. Additionally, CTX has been observed to permeate intact BBBs in both animal models and human brain tumors, with no cross-reactivity reported in non-malignant cells in the brain and other parts of the body [44]. Thus, CTX could be considered a promising targeting molecule for the development of novel diagnostic and therapeutic applications for GB and NB tumors.

The use of nanoparticle-based systems is attractive in biomedical research for the development of improved cancer-targeting diagnostic and therapeutic applications. Nanoparticles (NPs) improve the targeting of tumor cells by enhancing drug diffusion through the BBB and the specific targeting of diseased tumor cells, thereby limiting systemic toxicity [56,57]. NPs that are surface-functionalized with CTX as a targeting molecule, have been widely investigated and shown to demonstrate BBB-penetrating properties to reach GB tumors in vivo. These molecules are also useful for imaging-guided maximal surgical resection, drug delivery, and therapy monitoring [44,58,59,60,61]. Fluorescent-labeled CTX molecules and CTX-NP formulations for the delivery of chemotherapeutics and small interfering RNAs (siRNAs) have entered early clinical trials and preliminary results are promising [44,62,63,64,65]. 

This review summarizes part of a published thesis on this subject [66], highlights the pathogenesis and challenges associated with GB and NB cancers, and discusses the characteristics of CTX formulations as promising targeting peptides for these cancers. It also explores the mechanisms of action of CTX-based NPs, their common diagnostic and therapeutic applications for the management of GB, as well as potential application for the treatment of NB.

## 2. Glioblastoma Multiforme (GB): Standard Treatments and Challenges

Glioblastoma multiforme (GB) is the most highly invasive and aggressive, intracranial brain tumor diagnosed in adults, with a poor median survival time of 12–15 months and a 5-year survival rate of less than 7% [4]. There are many challenges associated with the treatment of GB, as highlighted in Figure 1, a major one being the blood-brain barrier (BBB) which diminishes the therapeutic value of most drugs for brain tumors due to its unique characteristics. GB is characterized by high inter-tumor and intra-tumor heterogeneity at cellular, molecular, histological, and clinical levels, resulting in poor and unchanged prognosis despite advancements in drug delivery strategies [67]. Standard treatment for GB follows the Stupp protocol which has been employed for the past two decades and involves maximal surgical resection, followed by radiotherapy and chemotherapy with temozolomide (TMZ) [68,69]. TMZ initiates DNA double-strand breaks, cell cycle arrest, and eventually cell death; however, it is associated with dose-limiting hematological toxicity [70]. Furthermore, TMZ is poorly soluble under physiological conditions and is prone to rapid hydrolysis which restricts its antitumor efficacy [71]. Drug resistance to TMZ is also often reported [72]. The size and anatomical location of GB tumors are major challenges to effective treatment as GB cells tend to overgrow rapidly, become highly invasive, and migrate deep into fragile brain regions, leading to incomplete tumor resection and tumor relapse [73]. In addition, a subpopulation of highly tumorigenic glioma stem cells (GSCs) with high plasticity and self-renewal properties add to tumor malignancy through their continued proliferation, invasion, stimulation of angiogenesis, reduction of anti-tumor immune responses and chemo-resistance [74]. Though not curative, extensive surgical resection is required to reduce the tumor size and relieve the intracranial pressure associated with GB symptoms and therefore presents a high risk for iatrogenic damage to healthy brain regions, leading to further complications [75]. The main disadvantage of radiotherapy is the non-specificity as normal cell DNA is also damaged, leading to permanent neuronal damage and radio-resistance of tumors and its attendant relapses following high dose radiation treatment or Radiotherapy utilizes X-rays, gamma rays or other charged particles to induce DNA even combination therapy [76]. Although the Stupp protocol may extend survival times, it does not cure GB, hence without treatment, the survival time is usually 3 months [77]. In 2015, a medical device based on tumor treating fields was introduced and applied on GB patients. However, this device did not significantly improve the median overall survival rates [68].

## 3. Neuroblastomas (NBs): Standard Treatments and Challenges

Neuroblastomas (NBs) are the most frequent extracranial solid brain tumors diagnosed in childhood and one of the major challenges in pediatric oncology, with a 5-year survival rate for patients presenting with high-risk NB tumors being below 40% [10]. A NB is an embryonal tumor of the sympathetic NS that arises because of disturbances within the migratory route of primitive neural crest cells along the sympathoadrenal lineage and normally originates in the adrenal glands or the paravertebral ganglia [78]. Thus, most tumors may be present in the neck, thorax, abdomen, and pelvis, as localized or metastatic tumors, while others may be secondary to such mental disorders as Hirschsprung’s disease or conditions such as congenital central hypoventilation syndrome and neurofibromatosis type 1 [79,80,81]. NB cells may invade other tissues and metastasize to bone marrow, bones, lymph nodes, skin, liver, lung, and brain [82,83]. Studies have shown that overexpression of the MYCN oncogene, which is known to be involved in embryogenesis, is one of the predominant factors implicated in NB [78]. On the other hand, the downregulation of the tyrosine kinase receptors (Trk), CD44, and overexpression of anaplastic lymphoma kinase (ALK) are the other molecular mechanisms that could lead to tumorigenesis or tumor expansion [84].

The standard treatment for NB consists of a coordinated sequence of chemotherapy, radiation therapy, surgical tumor resection, and combinations thereof, as well as myeloablative consolidation therapy with stem cell rescue and transplantation, 13-cis retinoic acid, and immunotherapy [85,86]. Surgery for low-risk NB may be sufficient support for chemotherapy with carboplatin, etoposide, cyclophosphamide, and doxorubicin [87]. However, surgical interventions are invasive, and incomplete tumor resection may require further chemotherapy, radiotherapy, and possibly stem cell transplantation [88]. For high-risk NB, long-term treatment with cisplatin, carboplatin, etoposide, vincristine, cyclophosphamide, and doxorubicin may be effective but causes systemic toxicity in the form of ototoxicity, thyroid function complications, cardiotoxicity, renal toxicity, future infertility complications and secondary malignancies [11,12,89,90]. The major long-term toxic side effects present as hearing loss, cognitive deficits, epilepsy, learning difficulties, endocrinopathies, growth reduction, thyroid function disorders, ovarian failure, and increased risks of secondary cancers [14,15,16,91,92]. Despite available treatment options, NB remains a major challenge in pediatric oncology and most survivors of high-risk NB often show spontaneous tumor regression after treatment [93], with more than half of the survivors not responding to high-dose chemotherapy and demonstrating multi-drug resistance [94]. The overall cure rate for high-risk NB is approximately 50% during the past two decades [10], and the lack of specificity of anticancer drugs to NB indicates that only low amounts of administered drugs can ultimately reach the tumor, with long-lasting side effects.

## 4. Current Challenges Associated with Drug Delivery to the Brain

A major challenge in the development of novel drugs for the treatment of NS tumors and other CNS diseases is the limitations posed by the BBB, which is the first line of defense from harmful substances in the blood that enters the brain circulation [95]. The combination of capillary endothelial cells held together by complex tight junction proteins, surrounding pericytes, the basal membrane, and astrocytic end-feet confers a high degree of selectivity to the BBB [96]. The BBB is approximately 200 nm thick, permitting the passage of small molecules (atomic mass < 400–600 Da) and hydrophilic molecules (atomic masses < 150 Da) via lipid-mediated diffusion, carrier-mediated transport systems, and receptor-mediated transport systems, while strictly preventing the paracellular entry of most chemotherapeutic drugs [97,98]. In addition, capillary endothelial cells in BBBs have a high concentration of drug efflux transporter proteins such as P-glycoprotein (P-gp) and multidrug resistance-associated proteins, resulting in reduced drug bioavailability [99]. 

As tumor cells invade the CNS and reach >0.2 mm^3^ of volume, the BBB is damaged and new blood vessels are formed through angiogenesis, leading to the formation of the blood-brain tumor barrier (BBTB) [7,100]. The newly formed capillaries are fenestrated, allowing the entry of approximately 12 nm-sized molecules through the BBTB (Sarin et al. [101]). With cancer progression and depletion of tight junction proteins, the capillary fenestrations become even more enlarged to allow for the passage of molecules of approximately 48 nm size and eventually 1 μm size, at which stage the BBB integrity is considered completely compromised [102]. The resultant leaky vasculature of most parts of the affected CNS tissue renders some chemotherapeutic drugs ineffective but peripheral areas of these tumors may have regions with an intact BBB resulting in the formation of favorable niches of cancer cell invasion and treatment resistance [103]. Thus, the combination of the BBB and the BBTB presents a unique challenge for brain tumor drug delivery.

The lack of specificity, poor drug delivery, drug resistance, and severe toxic side effects associated with standard treatments for GB and NB, are limitations to the effective management of these tumors. The avoidance of the systemic toxicity induced by chemotherapy and radiation treatments is crucial especially within the pediatric age group associated with NB patients because this can cause permanent changes to the developing body and increased risk of secondary cancers later in life [104]. Although there are many advances in research on GB and NB tumors in the past two decades, no effective treatments have been developed [69,94] as the current treatment options appear to be largely unsatisfactory, and inconsistent results have been reported for their effects on prolonging the median survival time of patients. In addition, significant remission is reported for early-diagnosed tumors but not for advanced disease stages [105,106].

Improvements in all treatment modalities are required for the successful management of cancers, and for GB and other CNS tumors, improved surgical resection techniques would result in fewer neurological side-effects and overall improvement in patient outcomes [107]. A better understanding of the exact mechanisms involved in drug delivery across dynamic biological barriers and the specific targeting of cancerous cells for treatment will foster novel and effective therapeutic strategies. Chlorotoxin (CTX) is one peptide that has recently generated interest in cancer research, especially for the targeted treatment of most CNS tumors [44,61,108], hence, the development of CTX-based nanoparticle treatments could offer promising outcomes for CNS tumors as discussed below.

## 5. Chlorotoxin (CTX): A Promising Natural Targeting Peptide for Cancers

In recent years, interest in exploiting the beneficial properties of venoms through the isolation of their peptides and investigating their efficacy as targeting molecules, have increased [109]. CTX is derived from the venom of the deathstalker scorpion (*Leiurus quinquestriatus*) and is a 36-amino acid peptide stabilized by four disulfide bonds, used as a potent targeting moiety due to its ability to bind to cancerous tissues, with high binding specificity for gliomas and NBs, and not to normal tissues [45,110]. CTX has emerged as a promising targeting agent for brain tumors due to its ability to specifically bind to 74 of the 79 World Health Organization (WHO) brain tumor classifications [110]. More than 15 normal human tissues have been shown to demonstrate negative CTX-binding properties [45,110]. 

CTX is considered safe and has been observed to permeate intact BBBs in both animal models and humans with brain tumors [111,112,113]. It is also a promising agent for the imaging and treatment of gliomas as demonstrated in clinical trials [64,65]. A synthetic CTX peptide labeled with ^131^I (commercial name ^131^I-TM-601) has already undergone early-phase clinical trials and received Food and Drug Administration (FDA) approval for a phase III trial in patients with newly diagnosed gliomas [114]. In addition to its high selectivity for targeting and binding of GB tumors, CTX has been shown to bind to a broad-spectrum of cancer cells including NB, medulloblastoma, breast cancer, ovarian cancer, prostate cancer, sarcoma, intestinal cancer, lung cancer and pancreatic cancer [45,48,49,50,51,53,54,62,110,115,116]. For peptides to be considered useful in therapeutics, they should normally possess the following characteristics: a small molecular size, clear activity on ion channels, and contain at least three disulfide bonds [60]. In addition, receptors present on cancer cells for these peptides should be uniquely or highly overexpressed in comparison to non-malignant cells, and a tumor-to-normal-cell expression ratio of 3:1 or higher is usually preferred to achieve the desired therapeutic effects [117]. Based on the above information, CTX meets all the important characteristics of a therapeutic peptide and is therefore a useful candidate in medical research considering its bioavailability and ability to induce target selectivity, which in turn reduces the side effects of drug resistance and systemic toxicity due to lack of specificity [44]. The selective binding of CTX to GB tumor cells has made its application as a targeting molecule for brain cancer therapy as well as a contrast agent for tumor optical imaging, very plausible [61].

### 5.1. Molecular Targets of CTX

The exact mechanisms by which CTX targeting occurs are not completely understood but potential primary cell surface targets have been identified over the years (Figure 2). Some studies have shown that CTX is an effective blocker of small conductance epithelial chloride channels [118,119] and mainly binds to overexpressed cancer cell surface receptors involved in the progression of tumors such as ClC-3 (chloride channel-3) in GB cells [37,38,110,119,120,121,122,123,124] which forms a protein complex with matrix metalloprotease-2 receptor (MMP-2) [48,49,50,116,125,126]; annexin A2 which is present in various cell lines [51,52] and has since been shown to be a potential target of CTX, and more recently estrogen receptor alpha (ERα) [53] and the Neuropilin-1 (NRP-1) [54,55] which is a vascular endothelial growth factor receptor responsible for tumor uptake. These molecules provide alternative methods for CTX targeting tumors in cancer diagnosis and therapy.

#### 5.1.1. Chloride Channels

Voltage-gated chloride (Cl^−^) channels have been associated with the proliferation and invasive migration of primary brain tumor cells [46,127]. Glioma cell shrinkage can be inhibited by Cl^−^ channel blockers leading to reduced invasion [45,46,127]. Intracellular Ca^2+^ was identified as a main regulator of cell motility due to Ca^2+^-activated ion channels [128] such as Ca^2+^-activated K^+^ channels which are known to elevate glioma migration [129]. Among the chloride channel protein family, chloride channel-2 and 3 (ClC-2 and ClC-3) are upregulated in glioma and are involved in the rapid changes in cell size and shape seen in dividing cells which invade extracellular brain spaces [130]. ClC-3 has been suggested to affect the invasion and migration of glioma cells by forming protein complexes with membrane type-I matrix metalloproteinase (MMP), MMP-2, tissue inhibitor of metalloprotein-2, and αvβ3 integrin, co-localizing with Ca^2+^-activated K^+^ channel to lipid raft domains of invadopodia [47,131].

The tumor-binding activity of a radioisotope ^125^I-labeled CTX (^125^I -CTX) was described by Soroceanu et al. [45] who showed its accumulation in tumor cells of GB-bearing mice, sparing normal neurons and astrocytes. Similarly, melanoma, neuroblastomas, medulloblastomas, and small-cell lung carcinomas in over 200 human surgical biopsy samples have shown CTX binding possibly because of their common neuro-ectodermal embryonic origin with glial cells [110]. A normal human brain and other tissues have also been shown to be consistently negative for CTX immunostaining [46].

The inhibitory effect of CTX on human GB-associated chloride channels was described by Ullrich et al. [120] and they also discovered the existence of specific CTX-sensitive glioma chloride currents in acute slices of human gliomas [123]. Cheng et al. [47] described the blocking activity of CTX on a single Cl-specific peptide blocker, a glioma-specific chloride channel (GCC) while Turner and Sontheimer [131] reported high-grade tumors expressing GCC more than low-grade tumors, while healthy tissues or tumors of non-glial origin poorly expressed these channels. GCC activity has also been suggested to regulate apoptosis and to be linked to changes in cellular cytoskeleton [122] as well as glioma cell morphology, proliferation, and migration [122,132]. In situ GCC expression using labeled CTX was found to correlate with the tumor grade, with only 40–45% of low-grade astrocytoma (WHO grade I–II) binding to it, versus 90% of high-grade tumors (WHO grade III) [45].

CTX was found not to inhibit volume-regulated, calcium-activated, and cyclic AMP-activated chloride channels expressed in various human, bovine, and monkey cells using concentrations of up to 1.2 µM [133]. However, Dalton and colleagues evaluated astrocytes found in injured adult rat brains and showed that CTX could inhibit calcium-activated chloride currents with an EC50 of ~50 nM [134]. It remains unclear if CTX may inhibit calcium-activated Cl^−^ channels; therefore, further research is required.

From the literature, it can be inferred that chloride channels may act as one of the markers of interest for targeting cancers, because of their role in tumor migration and growth, however, the findings reviewed above suggest the involvement of more than one type of chloride channel as GBs present with CTX being highly sensitive to ClC-3.

#### 5.1.2. Matrix Metalloproteinase-2 (MMP-2)

Matrix metalloproteinases (MMPs) are a family of calcium-dependent, zinc-containing endopeptidases, which are responsible for tissue remodeling and the degradation of the extracellular matrix (ECM), thus releasing several proteolytic and growth factors which contribute to tumorigenesis [135,136]. Thus, MMPs have invasive properties to tumor cells, regulate angiogenesis, trigger cell proliferation, and are upregulated in most cancer types, making them very important biomarkers for tumor detection [25]. High levels of MMP-2 and MMP-9 have been observed in patients with high-grade GB and high-risk NB and are associated with tumor progression [31,32,137,138,139,140]. MMP-2 appears to be a more promising molecular target of CTX [48], as its activation is a vital process required by GB for the degradation of the ECM during cell invasion and migration [141].

Although some researchers have suggested that MMP-1 plays a more important role than MMP-2 in the migration, remodeling, and invasiveness of GB [142], it has been shown that high levels of MMP-2 play a more important role in the virulent progression of cancer through its contribution to three vital processes: angiogenesis, metastasis, and invasion [25,143,144]. MMP-2 is specifically upregulated in gliomas as well as in other tumors of neuroectodermal origin such as NB, but not in the CNS [48,110]. In addition, MMP-2 expression is related to tumor aggressiveness and grade [122,130] and is reduced by CTX binding [48]. The reduced binding efficiency of CTX to GB cells in the presence of an MMP-2 inhibitor was demonstrated in a study by Veiseh et al. [115]. Jacoby et al. [114] proposed that CTX interacts with a cell surface protein complex that consists of MMP-2, membrane type-I MMP (MT1-MMP), a transmembrane inhibitor of metalloproteinase-2 as well as αvβ3, an integrin suggested to play an important role in angiogenesis and neural tumor invasion [145].

The structure of CTX is stabilized by 4 disulfide bonds and contains a β-sheet and helical structure. A computational study that predicted the binding of CTX with MMP-2 suggested that the β-sheet of CTX interacts in a region between the collagen-binding domain and the catalytic domain of MMP-2, whereas the α-helix of CTX does not appear to be involved in the interaction [146]. CTX has also been shown to inhibit MMP2 activity through fluorescence resonance energy transfer (FRET) substrate assay and gelatin zymography [147]. From the literature, it is proposed that ClC-3 and MMP-2 form a protein complex that is targeted by the CTX-peptide, and this action is thought to inhibit glioma cell migration and invasion through the induction of endocytosis of the MMP-2/ClC-3 protein complex [38,48,124]. Hence, CTX targeting MMP-2 has been widely investigated and proposed as one of the main molecular mechanisms for the development of CTX-based treatments for gliomas [48,49,50,53,112,114,115,116,125,148].

#### 5.1.3. Annexin A2

The Annexin protein family is a group of calcium-dependent phospholipid-binding proteins, involved in the repair of plasma membrane lesions triggered by different stimuli [149] as well as the control of various cellular functions including vesicle trafficking, vesicle fusion, and membrane segregation in a Ca^2+^-dependent manner through the binding of anionic phospholipids [150]. Other roles in cellular functions include angiogenesis, apoptosis, cell migration, proliferation, invasion, and cohesion [39,40,151]. In addition, annexins and their binding partners (the S100 proteins) are recognized regulators of the cellular actin cytoskeleton [152]. The surface protein annexin A2, a calcium-binding cytoskeletal protein localized on various tumor cells, has been shown to be a receptor for CTX on the surface of human cancer cell lines [51,52] and to be involved in cell migration, invasion, and adhesion [39,153,154,155]. Annexin A2 cell surface receptors have been implicated as molecular targets for CTX, based on studies on the effects of the commercially available CTX (TM-601) in human umbilical vein endothelial cells and human tumor cell lines [51]. The A2-complex comprises annexin-A2 and the protein p11, shown to be overexpressed on the surfaces of GB and is associated with poor prognosis [42,156]. TM-601 specifically binds to glioma cells but not normal brain tissues [157] and is found to bind to the surfaces of Panc-1 cells as well, depending on the level of annexin A2 expression [51]. A small interfering ribonucleic acid (siRNA) knockout of annexin A2 was found to result in reduced binding of a technetium-99m-labelled-TM601 in cell lines expressing annexin A2 [158]. A recent study demonstrated CTX binding to Hela cells known to overexpress Annexin A2 [159].

#### 5.1.4. Estrogen Receptor Alpha (ERα)-Mediated Signalling Pathway

Estrogen activates the estrogenic receptor (ER) signaling pathway and stimulates the expression of different genes that are involved in cell proliferation, causing breast cancer and related malignancies [160]. Studies have shown that ER can regulate the invasion and metastasis of tumor cells [161,162,163], hence, targeting ER signaling pathways is one of the important strategies for breast cancer treatment. A recent study by Wang et al. [53] found that CTX significantly inhibited breast cancer cell proliferation, migration, and invasion through binding to estrogen receptor alpha (ERα) to inhibit the expression of ERα, which inhibits the ERα/vasodilator-stimulated phosphoprotein (VASP) signaling pathway.

#### 5.1.5. Neuropilin-1 (NRP-1)

The most recent molecular target suggested for CTX is Neuropilin-1 (NRP-1), a vascular endothelial growth factor (VEGF) [54,55,164] known to be overexpressed in many cancers but naturally upregulated in normal lung and heart tissues [165,166,167,168]. Using nuclear magnetic resonance (NMR) spectroscopy and isothermal titration calorimetry (ITC), Sharma et al. [55] characterized the binding of CTX to the b1-domain of NRP1 (NRP1-b1) via a non-canonical primary sequence that satisfies the receptor binding motif through its tertiary fold. A novel peptide drug conjugate called ER-472, comprised of CTX linked to a cryptophycin analog, was found to possess antitumor activity related to NRP1 expression levels in xenografts, and its potency was significantly reduced following treatment with NRP-1 blocking antibodies or following knockout in tumor cells, confirming a role for NRP1-binding in ER-472 activity [54].

All the potential CTX molecular targets and receptors described above, appear to be over-expressed in diverse tumors, with MMP-2, Cl- channels, and Annexin A2 being the most widely investigated receptor targets; these are also known to be present in GB and NB. A recent study showed that neither CTX-CONH_2_ nor CTX-COOH affected cytotoxicity in a variety of Human tumor cell lines (U87MG, MCF-7, PC3, and A549) suggesting that terminal arginine amidation may not play an important role in the cytotoxic properties of CTX [169]. Other studies suggest that the C-terminal region plays a critical role in the bioactivity of CTX and inhibition of cancer growth and migration [54,114,170]. The exact mechanisms of CTX targeting action on cancer cells requires further investigation, perhaps through more detailed analysis that specifically identifies the structural determinants of CTX involved in binding to the respective potential receptors.

### 5.2. The Blood–Brain Barrier Crossing Potential of CTX

In addition to drug delivery for GB, the use of CTX as a carrier for delivering levodopa has been shown to result in increased distribution of dopamine in the brains of Parkinson’s disease mice [171]. Thus, CTX has been shown in several studies to demonstrate considerable potential for crossing the BBB to bind specifically to malignant brain tissue [59,60,61,108,172], and further diffuse deeply into the tumor environment, unlike other targeting agents such as antibodies [61,173]. CTX conjugated to fluorescent imaging agents and dyes such as Cy5.5 and 800CW were shown to bind to GB tumors in mice when delivered via tail injection [115,148].

Blaze Bioscience, Inc. has developed a fluorescent imaging agent composed of CTX covalently attached to the near-infrared fluorophore indocyanine green, commercially known as BLZ-100 (other names include: tozuleristide or Tumor Paint^®^) which is known to target tumor tissue for a complete and more precise surgical resection of brain tumors [65]. In addition, BLZ-100 demonstrated success in the preclinical resection of glioma [174], head and neck carcinoma [175], and soft-tissue sarcoma [176]. It has passed Phase I clinical trial and does not demonstrate any toxicity for doses up to 30 mg [64,65]. Presently, BLZ-100 is going through a joint Phase II/III trial for fluorescence-guided resection of pediatric CNS tumors (NCT03579602) [98]. The mechanism by which CTX crosses the BBB is not fully understood, however, Annexin A2 expression in BBB vascular endothelial cells has been suggested [51]. 

## 6. Nanotechnology for Cancer Applications

The development of novel diagnostic and therapeutic tools for the treatment of cancer requires innovations within the field of nanotechnology involving nanoparticles (NPs) (1–100 nm) which possess unique chemical, physical, and biological properties that render them attractive for biomedical applications, including in neuroscience research [177,178]. NPs that deliver therapeutic drugs along with an imaging moiety may provide multiple functions such as targeting, tracking, imaging, and treatment and are commonly referred to as “theranostic approaches” [179]. NPs are comprised of polymers, lipids, or metals, among other materials, that allow for encapsulation or surface conjugation with multiple therapeutic agents such as small molecules, peptides, or nucleic acids [57], with better therapeutic outcomes for many antitumor compounds [180]. Based on their sizes, NPs are naturally attracted to tumor sites with extensive abnormal angiogenesis, a phenomenon known as the enhanced permeability and retention effect (EPR) and often explored for passive targeting [181,182,183]. Passive targeting allows for the efficient localization of NPs within the tumor microenvironment, while active targeting facilitates the uptake of NPs by the tumor cells [184]. One main disadvantage of passive targeting is that it cannot be used for all tumors due to varied phenotypes [185], hence active targeting approaches are preferred.

In normal vasculature, endothelial junctions are ~5–10 nm in width but in tumor tissues, sizes of 100–780 nm have been reported, depending on the tumor type [186,187]. Thus, NPs of 15 > 50 nm diameter size easily cross the intact BBB [56,188] but in large and advanced brain tumors with extensive angiogenesis, the disrupted BBB allows NPs of size ranges of 5–200 nm to cross [56,188]. Other factors such as size, specificity for target sites, biocompatibility, stability in blood, evasion of the reticuloendothelial system (RES), site-specific drug release, etc. may also play a role [189]. Smaller NPs have larger surface areas which permit increased surface loading of therapeutic agents, while also promoting entry through tiny membrane passageways and increased drug bioavailability [190]. Thus, low doses or concentrations of therapeutic agents can be used, and systemic toxicity may be avoided [190]. The ideal size for maximum effect is 15–100 nm diameter as NPs below 10 nm can be cleared by the kidneys and those >150 nm will be removed by the RES [191,192], whereas NPs > 200 nm are usually considered undesirable for in vivo biomedical applications [193].

Active targeting of NP drug delivery systems in cancer therapy allows the drug effects to be specifically directed to cancer cells, facilitated by specific recognition binding sites that are either overexpressed on the surfaces of cancer cells or expressed at lower levels in normal cells [194]. Active targeting strategies have been accomplished by conjugating NPs with antibodies, peptides, and aptamers; however, for monoclonal antibodies (mAbs) that are generally used as targeting molecules for the targeted delivery of NPs, their large size, limited tissue penetration, cellular uptake, and conjugation difficulty to NPs, present significant challenges [195,196,197,198].

Peptides are considered more attractive targeting molecules based on their smaller size, lower immunogenicity, lower production costs, similar binding affinities to mAbs, and easier synthesis and modification methods [199,200,201]. In addition, peptides have a higher diversity, specificity, and targeting capability compared to other small molecule ligands [197,202,203]. Surface modification of NPs with synthetic polymers such as the FDA-approved polyethylene glycol (PEG) and other synthetic polymers such as polyvinyl alcohol (PVA), polyethyleneimine (PEI) or chitosan, act to enhance the solubility of hydrophobic materials and improve the biocompatibility of NPs through decreasing nonspecific binding and prolonging circulation durations in vivo [204,205]. These synthetic polymers also allow for the attachment of targeting molecules onto NPs for active targeting through the modification of terminal ends with various functional groups [204].

In recent years, biomimetic NPs have emerged as a promising drug delivery platform that enhances drug biocompatibility and specificity at the targeted site of disease, especially within the tumor microenvironment [206]. These NPs are inspired by nature and mimic the structure and function of biological molecules, such as proteins, enzymes, and lipids, enabling them to interact with biological systems in a manner similar to natural biomolecules, thereby facilitating diverse biomedical applications. Through modification with cell membranes to mimic biological functions associated with different cell membranes within biological systems. Researchers are focusing on constructing cell membrane-camouflaged NPs using a variety of cells, such as red blood cell membranes (RBCs), macrophages, and cancer cells. These cell membrane-camouflaged NPs inherit the composition of cell membranes, including specific receptors, antigens, and proteins that facilitate targeted drug delivery to tumors, immune evasion, and prolonged blood circulation times.

In nearly two decades, many CTX NP-based applications have proven to be novel diagnostic and targeting treatment agents for GB considering the many beneficial characteristics they possess, including their ability to penetrate the BBB, their high binding specificity for gliomas and other neuroectoderm-derived cancers, their ease of being internalized into tumor cells leading to prolonged retention time. Other reported characteristics to include their low toxicity or immunoreactivity profiles in human trials as well as the ease at which their structure can be modified to conjugate a variety of imaging or therapeutic agents without compromising the functionality of the peptide [61]. Thus, CTX-based NPs may be considered highly promising platforms for diagnostic imaging and targeted drug delivery for NS tumors.

## 7. CTX-NPs with Diagnostic Potential

Advances in nanotechnology innovation have resulted in the development of less invasive diagnostic and therapeutic approaches with high precision and specificity. Thus, many nano-based applications incorporate CTX to improve the visualization of GB tumors, as summarized in Table 1. Some of the CTX-conjugated NPs with diagnostic potential for GB have been used in magnetic resonance imaging (MRI), optical imaging, and single-photon emission computed tomography (SPECT) [59,60,61]. CTX-NPs delivered to target tumor tissues serve as MRI contrast molecules, while fluorophores or fluorescent probes that bind to molecular targets in tumors have been detected by optical imaging [61]. These techniques allow for precision-guided surgery without affecting normal tissues, based on the targeting function of CTX and the physicochemical characteristics of NPs. The conjugation of CTX to a fluorescent molecular probe, Cy5.5, described as “tumor r paint” was first used for intraoperative imaging [115] while a CTX functionalized iron oxide multifunctional nanoprobe (IONP-PEG-CTX) which could target glioma cells, was detectable by MRI [207] although superparamagnetic iron oxide NPs (SPIONPs) have now replaced these nanoprobes because they better enhance MRI. 

The application of CTX-functionalized SPIONPs for MRI/Optical imaging remains an area of active research [62,111,112,207,208,209,210,211]. Iron oxide NPs (IONPs) are composed of solid iron oxide cores (typically magnetite, Fe_3_O_4_, or its oxidized form maghemite, γ-Fe_2_O_3_) usually coated with synthetic polymers such as PEG, polyvinyl alcohol (PVA), polyethyleneimine (PEI), or chitosan to enhance the solubility of hydrophobic materials, limit the non-specific binding (thus prolonging circulation time), and enhancing tumor-specific targeting [212]. SPIONPs in a size range of 60–150 nm can possess different magnetic properties and functions differently in various applications [213]. Local interactions between iron and water protons accelerate the dephasing of protons to shorten transverse T2 relaxation times and enhance MRI contrast upon T2* imaging [214].

Some studies have shown that CTX functionalization onto the surface of IONPs using PEG or a copolymer of PEG and chitosan resulted in high targeting and the ability to cross the BBB [112,207,208,210,215]. The addition of Cy5.5 to CTX-IONPs in genetically engineered mice with no systemic toxicity was found to improve the targeting of glioma cells, the inhibition of glioma cells, easy crossing of the BBB, and prolonged detection of tumor cells by optical imaging and MRI [115,215,216,217]. The precise real-time detection of small foci of cancer cells with tumor margins could be achieved by optical imaging without affecting the BBB using CTX-NPs [208]. 

Fluorescence-based nano-imaging probes such as quantum dots (QDs) that provide excitation/emission wavelengths ranging from ultraviolet (UV) to near-infrared (NIR) light have also been used with CTX. QDs are composed of metals such as cadmium (Cd), zinc (Zn), selenium (Se), indium (In), and tellurium (Te), and have several significant advantages over fluorescent dyes and molecules (Jha et al [218]; Tarantini et al. [219]). QDs exhibit broad absorption and narrow emission spectra which makes them produce brighter emissions and have a higher signal-to-noise ratio compared with other fluorescent dyes [220], and are resistant to photo-bleaching [221]. Cadmium-free silver-indium-sulfide QDs conjugated to CTX [QD(Ag-In-S/ZnS)-CTX] were developed by Chen et al. [222] for cellular imaging studies and showed specific internalization into U87 human brain cancer cells while a stable polymer-blend dots CTX conjugate (PBdot-CTX) capable of crossing the BBB and specifically targeting tumor tissue in the ND2:SmoA1 medulloblastoma mouse model, was also developed [223]. The 15 nm PBdot-CTX conjugate was unaffected by photo-bleaching and was 15 times brighter than QDs [223]. The use of QDs may offer great advantages in experimental drug targeting and imaging but is limited for clinical use due to reported toxicity [224,225].

A class of NPs called up-converting NPs (UCNPs) have been reported as fluorescent imaging agents due to their ability to absorb low-energy near-infrared light (NIR) and “up-convert” to emit in the visible spectrum [226]. This characteristic allows tissue penetration of excitation light and minimizes auto-fluorescence, with the added benefit of photo-stability and prolonged fluorescing [227]. This allows UCNPs to be exploited for bio-imaging, bio-detection, and photodynamic therapy [228]. UCNPs composed of polyethyleneimine-coated hexagonal-phase thulium-doped sodium yttrium fluoride (NaYF(4):Yb), co-doped with erbium and cerium (NaYF4:Yb, Er/Ce) nanorods functionalized with CTX (PEI-NaYF(4):Yb, Er/Ce-CTX) have been shown to target C6 glioma-xenograft tumors in vivo without toxicity [226].

Deng et al. [229] showed that CTX-conjugated lanthanide-ion doped sodium gadolinium fluoride NPs (NaGdF4-Ho^3+^-CTX) demonstrated targeting towards glioma cells in vitro and in vivo, using MRI and fluorescence imaging techniques. Gu et al. [230] developed a glioma-targeted contrast agent by conjugating CTX to PEG-coated gadolinium oxide NPs (CTX-PEG-Gd_2_O_3_ NPs). The r1 value of CTX-PEG-Gd_2_O_3_ NPs (8.41 mM^−1^ s^−1^) was higher than that of commercially available Gd-DTPA (4.57 mM^−1^ s^−1^) and the enhancement of T1 contrast with a prolonged retention period up to 24 h within the brain glioma was observed due to CTX conjugation with low cytotoxicity. Similarly, europium-doped gadolinium oxide nanorods (Eu-Gd_2_O_3_ NRs) with paramagnetic and fluorescent properties were conjugated with doxorubicin (DOX) and CTX via PEGylation (CTX-PEG-Dox-Eu-Gd_2_O_3_ NRs) and found to target glioblastoma, deliver significant amounts of DOX to tumor sites and enhance MRI of the intracranial tumors in in vivo mouse models [231]. Dendrimer-based NPs are highly branched spherical structures that offer multifunctional applications in diagnosis and therapeutics [232]. Huang et al. [233] developed CTX-modified dendrimer-based conjugates that incorporated the MRI contrast molecule gadolinium (Gd(III)) which was composed of an L-lysine dendritic macromolecule conjugated to CTX either with Gd chelates or distyryl-substituted boradiazaindacene (BODIPY) fluorophore, resulting in enhanced uptake and retention time in tumor cells without toxicity. Many other CTX-dendrimer NPs have since been developed [211,234,235,236]. NIR fluorescent moieties are well suited for intraoperative CTX-based conjugates used for the identification of pre-malignant lesions and to improve the visualization of tumor boundaries. These moieties are poorly absorbed by water or hemoglobin and this decreases the interference from auto-fluorescence and optimizes signal intensity. Studies have shown that NIR fluorescent molecules modified with CTX such as Cy5.5 and IR Dye 800CW or indocyanine green (ICG) increased specificity and targeting with no impact on the efficacy of CTX for optical imaging [64,65,98,148,174,222,237]. A few studies have reported the use of NIR fluorescent molecules modified with CTX as well for MRI and other forms of imaging [209,216,217].

CTX has also been used in nuclear-based imaging techniques such as positron emission tomography (PET) and single photon emission computed tomography (SPECT) both of which have been exploited for dual imaging and treatment. Zhao et al. [238] first developed CTX multifunctional dendrimers labeled with radioactive ^131^I for SPECT imaging and radiotherapy of gliomas, followed by ^131^I-labeled CTX-functionalized gold NP entrapped in polyethylene naphthalate (poly(ethylene 2,6-naphthalate) (^131^I-labeled CTX- Au-PENPs) which was used as a nanoprobe for targeted SPECT/CT imaging in in vitro and in vivo radionuclide therapy of gliomas in a subcutaneous tumor model that also demonstrated BBB permeability [236,239]. Other theranostic NP formulations developed, include a polyethyleneimine (PEI), a methoxypolyethylene glycol (mPEG) CTX targeting, and a diethylenetriaminepentaacetic acid (DTPA) for ^99m^Tc radiolabeling DOX-loaded NPs (mPEI-CTX-^99m^Tc/DOX) [240]. These authors also found that the theranostic nano-complex demonstrated enhanced BBB permeability and tumor-targeting efficiency for gliomas using SPECT imaging and in vivo DOX drug delivery. CTX silver NPs (CTX-AgNP) were first studied in U87 human GB cell line [241] but a novel CTX-based polymeric NP radiolabeled with ^99m^Tc containing two cytotoxic agents, alisertib, and silver (Ag/Ali-PNPs-CTX-^99m^mTc), was later developed as a theranostic agent [242] and its targeting ability was tested on the U87 GB cell line and found to allow for in vivo visualization of bio-distribution in U87 tumor-bearing mice [242].

**Table 1 cancers-15-03388-t001:** Summary of CTX-NPs for diagnostic applications.

Name of Nanoparticle (NP) Formulation	Imaging Modality	Size in nm (Hydrodynamic Size/Core Size)	Ref.
mPEI-CTX-^99m^Tc/DOX	SPECT	394.77 nm	[240]
CTX-PEG-Dox-Eu-Gd_2_O_3_ NRs	MRI	116.3 nm	[231]
^131^I-labeled BmK-Au-PENPs	SPECT/CT	147 nm	[239]
^131^I-labeled CTX-Au-PENPs	SPECT/CT imaging	151 nm	[236]
Fe_3_O_4_;/PEG-FA–Cy5.5-CTX	MRI	<20 nm	[217]
^131^I-I-G5.NHAc-HPAO-(PEG-BmK CT)-(mPEG)	SPECT imaging	~4 nm	[211]
SPIONP-PEG-CTX	MRI	<100 nm	[243]
QD(Ag-In-S/ZnS)-CTX	Optical imaging	126 nm	[222]
Fe_3_O_4_/MnO–Cy5.5-CTX	MRI	25 nm	[216]
Ag/Ali-PNPs-CTX-^99m^Tc	Optical imaging	199 nm	[242]
NaGdF4-Ho^3+^-CTX	MRI/Optical imaging	44.2 nm	[229]
CTX-PEG-Gd_2_O_3_	MRI	3.46 nm	[230]
Pdot-CTX	Optical imaging	~15 nm	[223]
Gd-DTPA/BODIPY-dendrigraft poly-L-lysines-PEG-CTX	MRI	N/A	[233]
SPIONP-PEG-PEI-siRNA-CTX	Optical imaging	7.5 nm	[244]
IONP-PEG-Chitosan-DNA-CTX	MRI	48.8 nm	[62]
MFNP–CTX	MRI/Optical imaging	<100 nm	[210]
IONP-PEG-Chitosan-Cy5.5-CTX	MRI/Optical imaging	7 nm	[209]
PEI-NaYF(4):Yb, Er/Ce-CTX	Optical imaging	Width: 55 nm; length: 25 nm	[226]
NP-MTX-CTX	MRI	5–8 nm	[111]
IONP-PEG-CTX	MRI	10–15 nm	[208]
SPIONP-FITC-CTX	MRI/Optical imaging	80 nm	[245]
IONP-PEG-CTX	MRI/Optical imaging	10 nm	[207]

**Abbreviations**: MRI: Magnetic resonance imaging; SPECT: Single-photon emission computed tomography; CTX: Chlorotoxin; mPEI-CTX-^99m^Tc/DOX: methoxypolyethylene glycol (m), polyethyleneimine (PEI) ^99m^Tc radiolabeling NP loaded with doxorubicin (DOX); CTX-PEG-Dox-Eu-Gd2O3 NRs: Doxorubicin and CTX conjugated to polyethylene glycol coated gadolinium oxide NPs; ^131^I-labeled CTX-Au-PENPs: iodine-131 (^131^I-labeled) PEI-entrapped gold nanoparticles (Au PENPs) surfaced functionalized with CTX; Fe_3_O_4_/PEG-FA–Cy5.5-CTX: IONPs functionalized with polyethylene glycol and PEGylated folic acid (FA) labeled with Cy5.5 and CTX; ^131^I-I-G5.NHAc-HPAO-(PEG-BmK CT)-(mPEG): Bmk-CT: Buthus martensii Karsch CTX such as peptide conjugated to amine-terminated poly(amidoamine) dendrimers of generation 5 (G5.NHAc-HPAO), ^131^I-labeled; SPIONP-PEG-CTX: superparamagnetic iron oxide coated NPs with polyethylene glycol and CTX; QD(Ag-In-S/ZnS)-CTX: Cadmium-free silver-indium-sulfide Zinc shell (Ag-In-S/ZnS) Quantum dots functionalized with CTX; Ag/Ali-PNPs-CTX-^99m^Tc: Silver and alisertib polymeric NPs with ^99m^Tc radiolabeling and CTX surface functionalization; Fe_3_O_4_/MnO–Cy5.5: oleic acid-capped iron oxide manganese oxide with conjugation Cy5.5 dye and CTX; NaGdF4-Ho3+-CTX: Holmium doped d-sodium gadolinium fluoride (NaGdF4-Ho^3+^) nanoparticles conjugated with CTX; CTX-PEG-Gd_2_O_3_: CTX) to poly(ethylene glycol) (PEG) coated Gadolinium(III) oxide (Gd_2_O_3_)nanoparticles; Pdot-CTX: Polymer-blend dots conjugated with CTX; Gd-DTPA/BODIPY-dendrigraft poly-L-lysines-PEG-CTX: dendrigraft poly-L-lysines-PEG containing gadolinium ion diethylenetriamine pentaacetate NPs reacted with BODIPY dye; SPIONP-PEG-PEI-siRNA-CTX: superparamagnetic iron oxide NPs coated with polyethylene glycol and polyethyleneimine (PEI) conjugated with small/short interfering ribonucleic acid and CTX; IONP-PEG-Chitosan-DNA-CTX: iron oxide coated with polyethylene glycol and chitosan conjugated with deoxyribonucleic acid and CTX; MFNP–CTX: Magnetite and fluorescent silica nanoparticles functionalized with CTX; IONP-PEG-Chitosan-Cy5.5-CTX: iron oxide coated with polyethylene glycol and chitosan conjugated with fluorescent molecule Cy5.5-CTX; PEI-NaYF(4):Yb, Er/Ce-CTX: Polyethyleneimine-coated hexagonal-phase Ytterbium and Thulium Doped Sodium Yttrium Fluoride (NaYF(4):Yb), erbium and cerium co-doped nanoparticles; NP-MTX-CTX: IONPs conjugated to methotrexate (MTX), and CTX; SPIONP-FITC-CTX: superparamagnetic iron oxide NPs conjugated with fluorescein isothiocyanate (FITC) and CTX.

## 8. Therapeutic and Targeting Applications of CTX-NPs for GB Tumors

Many studies have reported the use of CTX-conjugated NPs and CTX-attached to fluorescent imaging agents for targeted precise surgical resection, and drug delivery of anti-cancer drugs/applications for the treatment of GB tumors and other tumors with no danger to normal cells [44,108]. Most of these formulations serve diagnostic, therapeutic, or theranostic functions in both in vitro and in vivo models of glioma as well as in clinical trials [44,61,64,65,108]. Table 2 provides a summary of CTX-based NP therapeutics used for the treatment of GB, but some applications overlap with diagnosis through the imaging techniques mentioned above, as seen in Figure 3. Many studies have shown that CTX-modified polymer or lipid-based NPs such as liposomes could be used as drug and gene delivery systems for glioma-targeted chemotherapy in brain tumors [113,246,247,248,249,250]. 

In gliomas, CTX inhibits the expression of MMP-2 and to achieve maximal inhibition, a dual system that employs an anti-cancer drug entrapped in or conjugated to a nano-carrier, together with the conjugation of CTX is used. Such a system makes use of the acidic environment inside the tumor environment to down-regulate MMP expression thus allowing for further treatment with chemotherapeutic agents [249]. In a study by Fang et al. [251], biocompatible polymer-coated IONPs conjugated to CTX or arginine-glycine-aspartic acid (RGD) were found to demonstrate that both NP-CTX and NP-RGD were target-specific to MMP-2 and αvβ3 integrin, respectively. Yue et al. [252] developed a transferrin receptor (TfR) monoclonal antibody (mAb) of rats (OX26) and CTX conjugated PEGylated liposome as a dual-targeting gene delivery system for GB which was found to significantly promote cell transfection, increase transportation of plasmid DNA across the BBB and target the brain glioma cells in vitro and in vivo. Qin et al. [246] demonstrated that CTX-liposomes specifically interact with MMP-2 present in brain cancer cells, which demonstrates targeting. Xiang et al. [113] first developed CTX-modified DOX-loaded liposomes (CTX-DoX-Lip) for glioma cells, but other studies have improved on this NP-based system for theranostic approaches by incorporating fluorescent molecules onto the liposomes in addition to CTX and chemotherapeutic drugs [171,248,250,253].

Other studies have reported on the encapsulation of small interfering RNAs in CTX liposomes [58,254] and antisense oligonucleotides [47,62] used as combination therapy for GB. CTX-functionalized NPs have been investigated for glioma gene therapy which has the potential to treat drug-resistant tissues, reduce unwanted toxicity to healthy cells, and provide a platform for therapy against multiple forms of cancer [255,256]. The first small interfering RNA (siRNA) magnetic nanovector (named NP-siRNA-CTX) with dual glioma targeting-specificity and dual therapeutic effect, was developed in 2010 for targeted cancer imaging and therapy [244]. These small 6–10 nm NPs demonstrated both increased small interfering RNA (siRNA) internalization by target tumor cells and intracellular trafficking towards enhanced knockdown of targeted gene expression. Mok et al. [257] reported that the multifunctional nanovector core coated with three different functional molecules [highly amine blocked PEI (PEIb), siRNA, and CTX] exhibited both significant cytotoxic and gene silencing effects for C6 glioma cells at acidic pH conditions, but not at physiological pH conditions. The NP-siRNA-CTX could also serve as an imaging tool for real-time monitoring of the delivery of therapeutic payload [244].

CTX has been functionalized to other noble metallic NPs such as silver (Ag) NPs and gold (Au) NPs and used for both detection and therapeutic applications (Table 1 and Table 2). Tamborini et al. [103] reported on AgNPs entrapped in Poly (lactic–co–glycolic acid) (PLGA) nanoparticles (PNP) conjugated to CTX (Ag-PNP-CTX). These NPs allowed the detection and quantification of cellular uptake by confocal microscopy, in both in vitro and in vivo experiments, and a higher uptake of Ag-PNP-CTX was reported in in vitro studies. Using a single whole-brain X-irradiation performed 20 h before NP injection, the expression of the CTX targets, MMP-2 and ClC-3 was enhanced as evidenced by the BBB permeabilization and increased internalization of Ag-PNP-CTX at the tumor site in vivo [103]. Locatelli et al. [241] first described CTX-functionalized on noble metallic NPs and developed a simple method for the synthesis of lipophilic AgNPs entrapped in a PEG-based polymeric NP conjugated with CTX (AgNPs-PNS-CTX). These NPs demonstrated significant cell-specific uptake in the U87 cell line in comparison to the Balb/3T3 cell line. The authors subsequently reported on the synthesis of multifunctional nanocomposites formed by polymeric NPs (PNPs) containing the anti-cancer drug alisertib, as well as AgNPs-conjugated with CTX and ^99m^Tc-radiolabeling (Ag/Ali-PNPs-CTX-^99m^Tc) (Table 1 and Table 2), which allowed significant tumor reduction as the result of *synergistic* effects of drug and NPs in U87 tumor-bearing mice [242]. The authors were also the first to later report on CTX and Cy5.5 functionalized gold nanorods (AuNRs-PNPs-Cltx/Cy5.5) for optoacoustic microscopy and photothermal therapy (PTT) using laser irradiation in U87 cells which consequently led to cell damage [258].

A recent study developed a nano drug delivery system composed of methoxypolyethylene glycol loaded with AuNPs, chemotherapeutic drug DOX and functionalized with CTX (mPEI-CTX/DOX). This product was found to have a higher IC_50_ value in human glioma cells than the free DOX, possibly due to the gradual release of the DOX from the mPEI-CTX/DOX NPs [240]. In addition to the MRI and fluorescence imaging properties of CTX-PEG-Dox-Eu-Gd_2_O_3_ NRs (Table 1 and Table 2), these NPs allowed for higher cytotoxicity in U251 human GB cells in vitro and no significant toxicity in HUVEC cells. In the in vivo experiments after tail-vein injection demonstrated no significant toxicity to normal organs, NPs accumulated in the brain tumors and appeared to inhibit tumor growth and metastasis [240]. Temozolomide (TMZ) has also been incorporated into CTX-NPs for improving target-specific drug delivery. A study by Fang et al. [71] reported on TMZ bound to chitosan-based NPs (NP-TMZ-CTX) exhibited higher stability at physiological pH, with a half-life 7-fold longer compared with free TMZ. Thus, the NP-TMZ-CTX was able to target GB cells and achieved 2–6-fold higher uptake and 50–90% reduction of half maximal inhibitory concentration (IC_50_) at 72 h post-treatment compared with NPs with TMZ but no CTX. 

Niosomes are nano-based drug delivery vesicles composed of non-ionic surfactants with or without cholesterol that are similar to liposomes, but could be synthesized smaller, are more stable, and are cheaper to manufacture in comparison to liposomes [259,260,261]. Niosomes coated with CTX and loaded with TMZ with an entrapment efficiency of 79.09 ± 1.56% were developed by De et al. [262] and found to have enhanced solubility and permeation into the brain in in vivo models due to CTX-conjugation, with less accumulation in other organs. TMZ drug resistance for GB is mediated by a DNA repair protein, O6-methylguanine-DNA methyltransferase (MGMT), which eliminates TMZ-induced DNA lesions [263]. Other studies report on combination treatments with small interfering RNA (siRNA)-based MGMT (siMGMT) inhibitors incorporated into CTX-NPs for targeting GB and sensitizing cells to TMZ for more effective therapeutic potential than free TMZ [264,265]. In another study, Mu et al. [266] developed a CTX-IONP conjugated with the drug gemcitabine (GEM) using hyaluronic acid (HA) as a cross-linker (IONP-HA-GEM-CTX) for GB therapy. This conjugate NP effectively killed GB cells without losing potency when compared to the free drug and showed a prolonged blood half-life and the ability to cross the BBB in wild-type mice [266]. Similarly, the chemotherapeutic agent, methotrexate (MTX) conjugated to CTX-NPs (NP–MTX– CTX) demonstrated increased uptake in 9 L rat glioma and significant cytotoxicity in tumor cells with prolonged retention of NPs observed within tumors in vivo [111]. Other studies by Agarwal et al. [249] showed that treatment with CTX-conjugated morusin-loaded PLGA NPs (PLGA–MOR–CTX) resulted in enhanced inhibitory effects and cell death in U87 and GI-1 glioma cells. The cytocompatibility observed with normal human neuronal cells (HCN-1A) together with enhanced lethal effects in GB cells, highlighted the potential of PLGA-MOR-CTX NPs as promising therapeutic nanocarriers for GB. In another study involving the drug sunitinib conjugated to CTX-coupled stable nucleic acid lipid NPs (CTX-SNALPs-miR-21 NPs), NPs showed preferential accumulation in brain tumors, promotion of efficient miR-21 silencing and enhanced antitumor activity, through decreased tumor cell proliferation, reduced tumor size as well as increased apoptosis activation [267].

Some earlier studies have reported on CTX-fluorescent NPs with effective targeting, BBB permeability, and high therapeutic effects both in vitro and in vivo [209,215,254]. Two recombinant versions of CTX named CTX-KRKRK-GFP-H6 and CTX-GFP-H6, were developed by Díaz et al. [159] and investigated in two Human cancer cell lines previously identified as targets for CTX, namely HeLa (overexpressing Annexin A2) and U87 (overexpressing MMP2). CTX-GFPH6 was found to have a significant cytotoxic effect on both cell lines, while CTX-KRKRK-GFP-H6 was more cytotoxic, and U87 cells were more sensitive than HeLa cells. In a recent study, a fluorescent nano-imaging agent (NIA) synthesized with polymalic acid with CTX, indocyanine green for fluorescence, and tri-leucin peptide for fluorescence enhancement (CTX-PMLA-LLL-ICG), was found to exhibit high specificity for U87 glioma cells [268]. This method involved the fluorescence-guided resection of GB using NIR light and has been shown to significantly improve the precision of tumor removal by 98% [268].

The efficacy of the respective NP conjugate products discussed above appears to be linked to apoptosis-mediated cell death mechanisms, possibly induced by CTX functionalization of the NPs. Wu et al. [269] reported on multifunctional Eu-doped Gd_2_O_3_ nanorods (Eu-Gd_2_O_3_ NRs) surface-functionalized with PEG to serve both as a hydrophilic coating and linkage molecule. This resulted in the covalent conjugation of the functional peptides RGD and CTX (RGD-Eu-Gd_2_O_3_ NRs-CTX) as a targeting nanovector for the detection and inhibition/therapy of early GB; these NPs could specifically target and adhere to U251 human GB cells, leading to cellular apoptosis. Pandey et al. [126] reported on a sophisticated multifunctional CTX-NP based on pH-responsive poly-l-lysine-coated Fe_3_O_4_@FePt core-shell NPs with CTX for mitochondrial targeted therapy of GB. The multifunctional NPs were efficiently localized inside mitochondria, induced oxidative stress by Fe, DNA strand breakage by Pt, and demonstrated the ability to disrupt mitochondrial function and induced apoptosis [126]. The authors also reported on effective PTT using NIR with these NPs [126].

**Table 2 cancers-15-03388-t002:** Summary of CTX-NPs for therapeutic applications.

Name	Therapeutic Effect	Theranostic Application	Size in nm (Hydrodynamic Size/Core Size)	Ref.
NP-siMGMT-CTX	The effective number of siRNAs (MGMT) delivered to tumors to sensitize both GB cells and GB stem-like cells (GSCs) to TMZ in vivo via CTX targeting	Yes	60.97 nm	[264]
CTX/DOTA/LND-PANPs Lf/CTX/TPP/DOTA/LND-PANPs	Increased localization of NPs in mitochondria both in vitro and in vivo, resulting in apoptosis. Photothermal therapy (PTT) with NPs occurred using NIR	Yes	<20 nm	[126]
mPEI-CTX-^99m^Tc/DOX	In vivo targeted delivery of DOX	Yes	394.77 nm	[239]
CTX-PEG-Dox-Eu-Gd_2_O_3_ NRs	No significant toxicity was reported in HUVEC cells, while toxicity was reported in U251 cells owing to CTX targeting MMP-2. In vivo experiments showed the inhibition of brain tumors with no significant toxicity to normal organs	Yes	116.3 nm	[230]
CTX-KRKRK-GFP-H6 and CTX-GFP-H6	Two recombinant CTX-fluorescent protein NPs demonstrated significant cytotoxicity in cell lines U87 (over-expressing MMP2) and Hela (overexpressing Annexin 2)	No	~12 nm	[267]
CTX-PMLA-LLL-ICG	Systemic IV injection into a xenogeneic mouse model carrying human U87 GB cells indicated tumor cell binding and internalization of NPs resulting in long-lasting tumor fluorescence which guided the resection of GB and significantly improved the precision of tumor removal	Yes	11.82 nm	[159]
CTX and mApoE-Dox-Lip	Enhanced DOX across the BBB via CTX-liposomes	No	184 nm	[249]
CTX-PLGA-Morusin	NPs resulted in enhanced inhibitory effects on U87 and GI-1 glioma cells	No	242.9 nm	[248]
CTX-TMZ noisome	Enhanced TMZ delivery into the brain in vivo with less deposition in the highly perfused organs	No	220 nm	[261]
M-CTX-Fc-L-Dox	Significant cytotoxicity observed with DOX loaded CTX- liposomes in U251 cells in vitro and tumor suppression in BALB/c mice bearing tumors of transplanted U251 cells in vivo	No	100–150 nm	[247]
RGD-Eu-Gd_2_O_3_ NRs-CTX	Nanorods specifically target U251 cells, leading to cellular apoptosis. In vivo results show NPs could effectively inhibit early tumor growth, without any damage to normal tissues/organ	Yes	~78 nm	[268]
IONP-HA-GEM-CTX	NPs effectively crossed BBB and killed GB cells, had prolonged blood circulation duration, and were excreted from the renal system	Yes	~32 nm	[265]
Ag-PNP-CTX	In vitro experiments performed with different human GB cell lines showed higher uptake of Ag-PNP-CTX, with respect to non-functionalized Ag-PNP NPs, and in vivo experiments showed that Ag-NP-CTX efficiently targets the tumors	Yes	199.1 nm	[103]
CTX-SNALPs-miR-21	MiRNA-21 silencing because of tumor-targeted CTX-NPs and decreased tumor cell proliferation and enhanced apoptosis in combination with Sunitinib	No	<190 nm	[266]
NP-TMZ-CTX	CTX-NPs demonstrated targeting of GB cells and 2–6-fold higher uptake and 50–90% reduction of IC_50_ at 72 h post-treatment as compared to NPs without CTX	Yes	<100 nm	[71]
Ag/Ali-PNPs-CTX-^99m^Tc	Significant tumor reduction was achieved in vivo as the result of the synergistic effects of Alisertib and NPs	Yes	199 nm	[241]
AuNRs-PNPs-Cltx/Cy5.5	NPs showed enhanced binding affinity toward GB cells in vitro using optoacoustic microscopy and PTT using laser irradiation of the cells led to cell damage	Yes	122.5 nm	[257]
CTX-Lip	CTX was attached to the surface of liposomes which interacts with the MMP-2 on the surface of U87 human glioma cell line cells and A549, demonstrating targeting	No	103.4 nm	[245]
CTX-IONP-siMGMT	Combination treatment of mice bearing orthotopic tumors with CTX-NP-siMGMT and TMZ led to a significant reduction of tumor growth	Yes	37.3 nm	[263]
CTX-SNALPs	Targeted NP-mediated miR-21 silencing in U87 and GL261 cells resulted in increased levels of the tumor suppressors PTEN and PDCD4, caspase 3/7 activation, and decreased tumor cell proliferation	No	<180 nm	[58]
AgNPs-PNS-CTX	Significantly higher uptake of Ag into U87 cells compared to the non-targeted NPs. Cytotoxic effect in glioma cell lines was also reported	No	130 nm	[240]
CTX-DoX-Lip	Increased cytotoxicity against U87 and U251 glioma and significant tumor growth inhibition in vivo	No	128 nm	[113]
NP-DNA-CTX	Enhanced uptake specifically into glioma cells in vivo	Yes	48.8 nm	[62]
IONPs-PEG-CTX and IONS-PEG-RDG	NP-CTX and NP-RGD were target-specific to integrin MMP-2 and αvβ3 integrin	Yes	~12 nm	[269]
NP(ION/PEG)-CTX-Cy5.5	NPs showed tumor-specific accumulation in vivo and no toxicity effects	Yes	13.5 nm	[270]
NP–CTX-chitosan-Cy5.5	Optimal serum half-life, biodistribution, stability, and non-toxicity were confirmed in mice	Yes	7 nm	[208]
MFNP-CTX	CTX-NPs demonstrated high specific cellular uptake	Yes	<100 nm	[209]
NP-siRNA-CTX	Increased small interfering RNA (siRNA) internalization by targeting glioma cells and intracellular trafficking towards enhanced knockdown of targeted gene expression	Yes	6–10 nm	[243]
NP-PEIb-siRNA-CTX	CTX-NPs showed long-term stability and good magnetic properties, significant cytotoxic effects, and gene silencing effects at acidic pH conditions for C6 glioma cells	Yes	~60 nm	[256]
NP-AF647-CTX-DNA	Results showed low cytotoxicity because of CTX targeting and excellent gene transfection efficiency	Yes	134.8 nm	[253]
NP-CTX-AF680	The NPs enhanced cellular uptake via MMP-2	Yes	~11 nm	[112]
NPCP-Cy5.5-CTX	NPs showed cytotoxicity, sustained retention in tumors, and the ability to cross the BBB and specifically target brain tumors in vivo	Yes	33 nm	[214]
NP-MTX-CTX	Increased cytotoxicity of methotrexate (MTX) in GB cells and prolonged retention of NPs was observed within tumors in vivo NPs	Yes	5–8 nm	[111]

**Abbreviations**: NP: nanoparticles; CTX: Chlorotoxin; NP-siMGMT-CTX:IONPS small interfering RNA (siRNA)-based MGMT (siMGMT) inhibitors and CTX conjugated to IONPs; CTX/DOTA/LND-PANPs Lf/CTX/TPP/DOTA/LND-PANPs: pH responsive poly-l-lysine coated Fe_3_O_4_@FePt core shell NPs with CTX for mitochondria targeted (Mito-PANPs); mPEI-CTX-^99m^Tc/DOX: methoxypolyethylene glycol (m), polyethyleneimine (PEI) ^99m^Tc radiolabelling NP loaded with doxorubicin (DOX); CTX-PEG-Dox-Eu-Gd_2_O_3_ NRs: Doxorubicin and CTX conjugated to polyethylene glycol (PEG) coated gadolinium oxide NPs; CTX-KRKRK-GFP-H6 and CTX-GFP-H6: 2 different fluorescent protein NPs named CTX-KRKRK-GFP-H6 and CTX-GFP-H6 conjugated to; CTX-PMLA-LLL-ICG: Polymalic acid (PMLA) conjugated with CTX, tri-leucine peptide (LLL) and indocyanine green (ICG); CTX and mApoE-Dox-Lip: liposomes entrapped with DOX and dually functionalized with ApoE-derived peptide (mApoE) and CTX; CTX-TMZ noisome: Noisome entrapping Temozolomide (TMZ) surface functionalized with CTX; RGD-Eu-Gd_2_O_3_-CTX coated europium dopped gadolinium oxide nanorods, functionalized with arginine-glycine-aspartic acid (RGD) and CTX; CTX-PLGA-Morusin: CTX conjugated to poly(lactic-co-glycolic acid) (PLGA) NPs loaded with morusin; M-CTX-Fc-L-Dox: liposome loaded with DOX and modified with CTX fused to human IgG Fc domain without hinge region in monomeric form (M-CTX-Fc); IONP-HA-GEM-CTX: Iron oxide NPs conjugated with chemotherapeutic drug gemcitabine (GEM) and CTX using hyaluronic acid (HA) as a crosslinker; Ag-PNP-CTX: silver NPs (AgNPs) entrapped in poly(lactic-co-glycolic acid) (PLGA) nanoparticles (PNP) conjugated to chlorotoxin (CTX); AuNRs-PNPs-Cltx/Cy5.5: Gold nanorods entrapped in poly(lactic-co-glycolic acid) (PLGA) nanoparticles (PNP) conjugated to CTX and Cy5.5; CTX-SNALPs-miR-21: CTX-coupled stable nucleic acid lipid particles (SNALPs) for miR-21 silencing; NP-TMZ-CTX: chitosan-based NPs with TMZ, conjugated with CTX; NP:DNA-CTX: IONPS coated with PEG and PEI, DNA was encapsulated into NP and CTX was conjugated on the surface; Ag/Ali-PNPs-CTX-^99m^Tc: Silver and alisertib polymeric NPs with ^99m^Tc radiolabelling and CTX surface functionalization; AgNPs-PNS-CTX: Silver polymeric NPs with CTX surface functionalization; CTX-Lip: CTX liposomes; CTX-IONP-siMGMT: small interfering RNA (siRNA)-based MGMT (siMGMT) inhibitors incorporated into CTX-IONPs; CTX-SNALPs: CTX-coupled stable nucleic acid lipid particles; CTX-DoX-Lip: CTX functionalized liposomes entrapping DOX; NP-DNA-CTX: IONPS conjugated with DNA and CTX; IONPs-PEG-CTX and IONS-PEG-RDG: IONPs coated with PEG conjugated with CTX or RDG; NP(ION/PEG)-CTX-Cy5.5: IONPs surface functionalized with CTX and Cy5.5; NP-CTX-chitosan-Cy5.5: IONPs coated with chitosan and conjugated to CTX and Cy5.5; MFNP-CTX: Magnetite and fluorescent silica nanoparticles functionalized with CTX; NP-siRNA-CTX: IONPS coated with PEG and conjugated to siRNA and CTX; NP-PEIb-siRNA-CTX: IONPS coated with polyethylene glycol (PEG)-grafted chitosan, and polyethyleneimine (PEI) with polyethylenimine (PEI) and conjugated to siRNA and CTX; NP-AF647-CTX-DNA: IONPS conjugated with Alexa Fluor 647 dye (AF647) and DNA; CTX; NP-CTX-AF680: IONPS conjugated with Alexa Fluor 680 dye (AF680); NPCP-Cy5.5-CTX: PEGylated-chitosan branched copolymer (CP) NPs conjugated with Cy5.5 and CTX.

## 9. Prospective Applications of CTX-NP Formulations

### 9.1. Optoacoustic Imaging Using CTX-NPs

CTX-NPs could also be used in other diagnosis and treatment applications for both GB and NB. For instance, optoacoustic imaging is one area of interest that has been investigated preclinically and involves the use of acoustic emissions from pulsed light energy to visualize biological structures at high optical contrast and acoustical resolution [271]. Commonly used acoustic imaging contrast agents are microbubbles (MBs), nanobubbles (NBs), and nanodroplets (NDs) that can be used with photo-acoustic and ultrasound imaging [272]. Stable oscillations of MBs are caused by exposure to low acoustic pressure, a process termed stable cavitation [273]. MBs were initially developed as diagnostic ultrasound contrast agents but have since been explored for targeted drug delivery by enhancing vascular permeability through cavitation when bubbles occur in ultrasound fields [274]. MBs may have difficulty in penetrating the deep tissue layers, whereas NBs hold the potential for extensive delivery into tissues through blood vessels and NDs can pass through the leaky microvasculature and reach the perivascular space, such as a tumor’s interstitial space [273]. Modifications of bubble surfaces allow the targeting of diseased tissues, reduced immunogenicity, and prolonged circulation times. Various bubble formulations are used for ultrasound imaging [275] targeted drug delivery [276,277,278,279], gene delivery [280,281], and hyperthermia treatment [282], however, research in this field incorporating CTX as a targeting molecule has not been explored but has been previously suggested as a promising diagnostic application for GB [61], and should also be considered for NB.

### 9.2. Diagnostic and Therapeutic Potential of Biomimetic CTX-NPs

Combining cell membrane-derived biological functions and NPs has allowed biomimetic NPs to be developed for numerous applications in tumor imaging techniques such as CT imaging, MRI, fluorescence imaging, and photoacoustic imaging [283]. To further improve the accumulation of chemotherapeutic agents and anti-cancer molecules at glioma locations, active-targeting biomimetic liposomes have gained momentum in neuroscience research. Li et al. [284] prepared elemene (ELE) and cabazitaxel (CTX) liposomes conjugated with transferrin (Tf) and embedded with the cell membrane proteins of RG2 glioma cells into liposomes (active-targeting biomimetic liposomes, Tf-ELE/CTX@BLIP). These NPs produced excellent BBB permeating capacities, highly significant homologous targeting and immune evasion in vitro, and a 5.83-fold intake rate compared with liposomes without Tf and cell membranes of RG2 cells. Based on the observation of elevated lactate (LA) in resected GB, Lu et al. [285] developed biomimetic therapeutic NPs that deliver agents for LA metabolism-based synergistic therapy. These NPs were encapsulated in membranes derived from U251 GB cells that readily penetrated the BBB and targeted GB through homotypic recognition. After reaching the tumors, lactate oxidase in the NPs converts LA into pyruvic acid (PA) and hydrogen peroxide (H_2_O_2_). The PA inhibits cancer cell growth by blocking histones expression and inducing cell-cycle arrest. In parallel, the H_2_O_2_ reacts with the delivered bis [2,4,5-trichloro-6-(pentyloxycarbonyl) phenyl] oxalate to release energy, which is used by the co-delivered photosensitizer chlorin e6 for the generation of cytotoxic singlet oxygen to kill glioma cells. Such a synergism ensures strong therapeutic effects against both glioma cell-line-derived and patient-derived xenograft models. A recent study demonstrated biomimetic Dp44mT-NPs selectively-induced apoptosis in Cu-loaded GB which resulted in potent growth inhibition [286]. Biomimetic NPs can also be a promising phototheranostic nanoplatform for brain-tumor-specific imaging and therapy. By embedding glioma cell membrane proteins into NPs, Jia et al. [287] successfully synthesized biomimetic ICG-loaded liposome (BLIPO-ICG) NPs which could cross BBB and actively reach glioma at the early stage due to their specific binding to glioma cells as a result of their excellent homotypic targeting and immune escaping characteristics. High accumulation in the brain tumor with a signal-to-background ratio of 8.4 was obtained at 12 h post-injection. At this time point, the glioma and its margin were visualized by NIR fluorescence imaging. Under imaging guidance, the glioma tissue was completely removed and in addition, after NIR laser irradiation (1 W/cm^2^, 5 min), the photothermal effect exerted by BLIPO-ICG NPs efficiently suppressed glioma cell proliferation with a 94.2% tumor growth inhibition. A novel “cocktail therapy” strategy based on excess natural killer cell-derived exosomes (NKEXOs) in combination with biomimetic core–shell NPs was developed for tumor-targeted therapy in CHLA-255 cells NB cells [288]. The NPs were self-assembled with a dendrimer core loading therapeutic miRNA and a hydrophilic NKEXO shell. NKEXO NP cocktail showed highly efficient targeting and therapeutic miRNA delivery to NB cells in vivo, as demonstrated by two-photon excited scanning fluorescence imaging (TPEFI) and with an IVIS Spectrum in vivo imaging system (IVIS), leading to dual inhibition of tumor growth. The authors proposed this NP cocktail as a new strategy for tumor therapy. Despite the rapid development in this field, not much has progressed to the clinical stage and there are no CTX biomimetic-based NPs to produce a highly specific and effective drug delivery for GB and NB which can be used for clinical purposes, therefore more research in this area is required.

### 9.3. Hyperthermia Treatment Using CTX-NPs

So far, the therapeutic applications of CTX have focused on conjugating the peptide to NPs to allow for targeted delivery of drugs and therapeutic agents or the visualization of tumors or both, with very few applications involving hyperthermia treatment (HPT) [126,258], which is one of the oldest treatments for cancer and a promising minimally invasive thermal therapy [289]. This is an effective treatment modality that utilizes heat energy to destroy cancer cells that are more prone to generate heat, owing to their overall increased metabolic rates [290]. A prospective hyperthermia treatment application of CTX-NPs for GB and NB is to induce intracellular heat stress with the use of NPs (at a temperature range of 41–47 °C), resulting in mitochondrial swelling, protein denaturation, alteration in signal transduction, cell rupturing and induction of necrosis/apoptosis [290,291,292]. Some of the common drawbacks of hyperthermia treatment include its invasiveness, incomplete tumor destruction, low heat penetration in the tumor region (lesions > 4–5 cm in diameter), excessive heating of surrounding healthy tissues (non-specificity), thermal under-dosage in the target region, heat dissipation by the blood as well as the development of thermotolerance [290,293]. The use of magnetic and metallic NPs (MNPs) to induce localized NP-mediated hyperthermia within cancer cells, as illustrated in Figure 4, has recently gained considerable interest in cancer nanotechnology research but this has yet to be fully exploited for the brain, and other CNS tumors. Recent studies have reported on the promise of deep intracranial thermotherapy with MNPs for brain tumors [294,295,296] with some entering clinical trials [297]. In general, both whole-body and regional hyperthermia treatments result in poor tumor specificity and constitute a strong limitation to the clinical application of this technique [290,293].

Some of the most explored magnetic NPs for HPT based on their superior magnetic properties include iron, cobalt, nickel, manganese, zinc, and gadolinium, as well as their alloys and oxides—CoFe_2_O_4_, NiFe_2_O_4_, ZnFe_2_O_4_, CuFe_2_O_4_, MnFe_2_O_4_, Gd-doped Zn-Mn and Zn-Mn-doped iron oxides [298,299,300,301,302,303,304,305]. However, the use of most of these metals and alloys is mostly limited by potential toxicity and chemical instability [306]. Interestingly, IONPs have excellent self-healing properties and have been licensed for use in clinical applications by the FDA and the European Medicines Agency (EMA) [307]. Although IONPs have been licensed for use in clinical applications by the FDA, these NPs have been reported to exhibit toxicity in vitro and in vivo. IONPs can cause toxicity to cells by inducing oxidative stress in cells and affecting the cell surface roughness which could also change the shape and alter the response by the cellular cytoskeleton [308]. Ultra-small IONPs showed high toxicity in vivo due to the distinctive capability in inducing the generation of reactive oxygen species (ROS), and ferroptosis based on Fe^2+^ and radicals (OH) in multiple organs, especially in the heart [309]. The toxicity is dependent on the size and iron element. External alternating magnetic field (AMF) is used with IONPs/SPIONPs to produce heat energy for the thermal ablation of cancer cells [310] in controlled environments [311]. Increasing the strength of the AMF field may result in inductive tissue heating from eddy current losses, which is independent of the presence of IONPs/SIONs; this may restrict the extent to which the AMF field can be increased [312,313].

Noble metals are excellent conductors of thermal energy that offer a non-invasive and effective therapeutic strategy for intracellular hyperthermia [314]. AuNPs and platinum NPs (PtNPs) have strong local surface plasmon resonance (SPR) effects, hence, upon exposure to light, can absorb sufficient photon energy to generate photothermal properties [315]. AuNPs and PtNPs have been used in both in vivo and in vitro studies to demonstrate photothermal therapy (PTT)-induced cytotoxicity through exposure to near infra-red (NIR) light (650–950 nm) using special lasers [316,317,318]. Interest in bimetallic NPs as anti-cancer applications has increased due to their value in enhancing drug delivery strategies and NP-mediated hyperthermia treatments [319]. A number of studies reported that bimetallic gold-platinum NPs (AuPtNPs) of different sizes and shapes exhibit better photothermal effects and higher radiation-enhancing properties than the respective monometallic NPs (AuNPs and PtNPs), possibly due to the synergistic effects of the two composite metallic atoms and new surface properties that are different in their monometallic NPs [320,321,322,323,324,325,326,327]. Graphene quantum dots (GQDs) are considered promising nanomaterials for the PTT of cancer due to their biocompatibility, capability of crossing biological barriers, and rapid excretion as a result of their small size [328,329]. In a recent study by Perini et al. [330], GQDs in combination with DOX and TMZ were tested on a complex 3D spheroid model of GB. They combined GQDs-mediated PTT and chemotherapy at subtherapeutic doses on GB spheroids and observed a significant reduction both in spheroid growth and viability in the time-span of two weeks, along with a considerably higher penetration depth and uptake of the antitumor drug inside the GB model. Their findings suggested that GQDs could increase membrane permeability through PTT conversion in a reliable tumor model [330]. Lin et al. [331], developed GQDs, which were conjugated with antibodies against GD2, a disialoganglioside, and a surface antigen expressed on NB cells, to become anti-GD2/GQDs. The efficiency of targeting and imaging of anti-GD2/GQDs were investigated in NB cells and the authors found that there was significant accumulation of the fluorescence of anti-GD2/GQDs in NB cells both in vitro and in vivo. GQDs may potentially be used for the targeting and imaging of GB and NBs through surface functionalization with CTX.

MXenes are a new class of two-dimensional (2D) nanomaterials made of transition metal carbides, nitrides, and carbonitrides. Since 2011, they have been attracting attention due to their unique combination of electrical and mechanical properties, as well as their hydrophilicity. The potential applications of MXenes in nanomedicine are numerous, such as sensors, antibacterial agents, targeted drug delivery, cancer photo/chemotherapy, tissue engineering, bioimaging, and environmental applications, including sensors and adsorbents [332]. MXene quantum dots are produced by exfoliating MXene sheets into ultrathin nanosheets and then processing them into nanocrystals. These nanocrystals are typically only a few nanometers in size and exhibit quantum confinement effects due to their small size, leading to unique electronic and optical properties. Nitride-based MXene and titanium nitride quantum dots (Ti2N QDs) was produced by Shao et al. [333]. A dose of 80 μg mL^−1^ Ti_2_N QDs showed no cytotoxicity to U87 cells. However, supplementing this treatment with NIR laser irradiation for 5 min led to almost all of the cells being killed. These studies suggested a tremendous potential for the use of MXenes in cancer treatment for GB. A recent study by Zhang et al. [334] reported on the fabrication of 2D nano-sonosensitizers/nanocatalysts (Ti_3_C_2_/CuO_2_@BSA) for the in situ generation of nano-sonosensitizers by responding to the tumor microenvironment, achieving the high-performance and synergistic sonodynamic (SDT)/chemodynamic tumor therapy. SDT utilizes a tumor-localizing sonosensitizing agent (NP) which is activated by ultrasound and produces greatly ROS to destroy tumor cells [335]. CuO_2_ NP integration on 2D Ti_3_C_2_ MXene achieved in situ H_2_O_2_ generation in an acidic tumor microenvironment for oxidizing Ti_3_C_2_ to produce TiO_2_ nano-sonosensitizers, accompanied by the enhanced separation of electrons (e^−^) and holes (h^+^) by the carbon matrix after oxidation, further augmenting the SDT efficacy. Ultrasound irradiation during the sonodynamic process also enhanced the Cu-initiated Fenton-like reaction to produce more ROS for synergizing the sonodynamic tumor therapy. The experimental results confirmed and demonstrated the synergistic therapeutic effects of chemodynamic and sonodynamic nanotherapy both in vitro and in vivo. Currently, there are no investigations with CTX for applications with MXenes, which is an area of research that needs to be explored. 

The use of NIR PTT is limited to subcutaneous/superficial malignant tumors because of minimal tissue penetration (~3 cm depth) by NIR light which may not be suitable for deep-seated brain tumors [336]. Hence, other applications such as external radiofrequency (RF) ablation are suggested as radio wave energy has been shown to penetrate more deeply located tumors than NIR light [337,338,339,340]. At 220 MHz, RF penetration is 7 cm and increases with a decrease in frequency, whereas RF penetration is 17 cm at 85 MHz [341,342]. Radio waves are safe, low-frequency electromagnetic waves with low tissue-specific absorption rates (SAR) and are therefore excellent for applications involving whole-body tissue penetration [337]. The heating properties of AuNPs and PtNPs have been investigated using RF currents and shown to offer some promise for non-invasive RF anti-cancer therapy; however, reports on targeted bimetallic AuPtNPs for this application are limited in the literature [343,344,345]. CTX-conjugated AuPtNPs and other bimetallic NPs need further investigation as potential heating agents for RF-based hyperthermia for the treatment of such deep-seated tumors as GB, as they target only tumor cells with minimal adverse effects on surrounding healthy cells.

Hyperthermia treatments are also known to sensitize cells to other forms of standard therapy, including radiation and chemotherapy having potential in combination treatments [293,346,347]. Another example of the use of nanoparticle-mediated hyperthermia treatments is in thermosensitive controlled drug release [318,348,349]. This concept was recently explored by Pandey et al. [126] using CTX-functionalized bimetallic NPs (Table 2) for mitochondria targeting and chemo-photothermal therapy with NIR. Research into multimodal CTX-NPs incorporating RF-hyperthermia for GB and NB treatment is required as this may yield results that could offer new hope for the effective treatment and management of these tumors.

## 10. CTX-like Peptides

Another group of molecules with prospective applications for targeted cancer diagnosis and therapy is the “CTX-like peptides”. Since the discovery of CTX from the venom of the *Leiurus quinquestriatus* scorpion, a few CTX-like peptides with similar primary features and functions as CTX, have been isolated and identified [59,61,65,118,350,351]. CTX-like peptides are considered ion channel blockers and MMP-2 inhibitors because they interact with MMP-2 on cell membrane surfaces, resulting in anti-metastasis or antitumor effects with minimal-to-no effects on normal cells [173]. Other scorpion venom peptides with similar primary structure as CTX include AaCTX, ClTx-a, -b, -c, -d, BmKCTa, BmKCL1, Lqh-8:6, Be I5A, BeI1, AmmP2, GaTx1 and GaTx2 [59,352,353]. AaCTX isolated from *Androctonus australis* scorpion, has 61% identity with CTX and was suggested to have inhibitory effects on invasion and migration through chloride channels [354]. Sequence alignment showed that BmKCTa (isolated from *Buthus martensii Karsch* venom), GaTx1, and GaTx2 (isolated from *Leiurus quinquestriatus* venom) have 67%, 64%, and 38% similarity with CTX respectively, and show some activity on chloride and other ion channels [44,59,60,355,356]. GaTx1 is a highly specific blocker for the cystic fibrosis transmembrane conductance regulator (CFTR) channel, a receptor belonging to the ABC family, with intrinsic Cl^−^ channel activity [357].

ClC-2, another member of the ABC family of chloride channels like ClC-3, is also upregulated on the surfaces of glioma cells, but its physiological role is not completely understood; it has been suggested to play a similar role as ClC-3 in glioma cell invasion, and migration [124]. GaTx2 inhibits ClC-2 by slowing down its activation [358], and the resulting inhibition is voltage-dependent. BmKCTa, the most common CTX-like peptide investigated, also demonstrated the inhibition of glioma cell proliferation, migration, and invasion in a fashion similar to CTX with MMP-2 as the potential target [211,235,316,351,355,359,360,361]. The CTX-like peptide, Bs-Tx7, from the venom of *Buthus sindicus* scorpion, has a scissile peptide bond (i.e., Gly-Ile) for MMP2 and demonstrated 66% sequence identity with CTX and 82% sequence identity with GaTx1 [362]. In another study, Xu et al. [363] identified the CTX-derivatives CA4 and CTX-23, which showed high selective binding to malignant glioma cells and inhibited rodent and human glioma cell growth at low concentrations, with minimal-to-no toxicity to primary astrocytes and neurons. Furthermore, these authors also found that CA4 and CTX could normalize tumor vessel morphology and vessel density in the peritumoral brain areas [363]. Thus, more research is required to understand the specific mechanisms of action of these CTX-like peptides, as well as the plausibility of their use as potential targeting agents for the treatment of GB and NB tumors.

## 11. Conclusions and Future Directions

The rising incidence of GB and NB imposes major global health challenges, with a substantial economic burden for patients, health insurance providers, and health authorities alike. The pathophysiology of these tumors involves the elevation of many surface proteins such as MMPs which contribute to proliferation and metastasis. Therefore, strategies that inhibit the over-expression of MMPs may reduce cancer progression. CTX is a peptide that holds great promise for use as a theranostic agent for NB, GB, and other solid tumors, with many CTX-NPs applications constantly being investigated. CTX easily penetrates the BBB, has a high binding affinity for gliomas and other cancers including NB, but not normal tissues, and is reported to be readily retained for longer periods in cancer tissue with little or no toxicity or immunoreactivity. There is substantial evidence to show that the efficacy of CTX is related to its ability to cross the BBB as well as its high tumor-binding function mediated by the molecular targets namely, chloride channels, MMP-2, annexin A2, and recently, ERα and NRP-1. However, more research is required to fully elucidate the mechanisms involved in the binding of CTX to tumor molecular targets as well as its crossing of the BBB. Incorporating CTX onto NPs such as biomimetic NPs, GQDs, and MXenes, as well as applications for Optoacoustic imaging using CTX-NPs are all areas that require more research. CTX-Noble bimetallic NPs have recently demonstrated superior anti-cancer activity when compared to monometallic NPs, especially for hyperthermia-based treatments; however, only a few studies have reported on CTX functionalized NPs and bimetallic NPs for hyperthermia treatments, thus requiring further investigation. Finally, only a few studies have reported on the use of CTX-NPs in NIR photothermal therapy, and to the best of our knowledge, no radiofrequency-based hyperthermia studies involving CTX-NPs exist in the literature, necessitating more studies on these applications, since they may be highly advantageous for deep-seated tumors such as GB. Overall, this review highlights the potential of CTX and CTX-NPs as safe and effective diagnostic and therapeutic applications for GB and NB tumors.

## Figures and Tables

**Figure 1 cancers-15-03388-f001:**
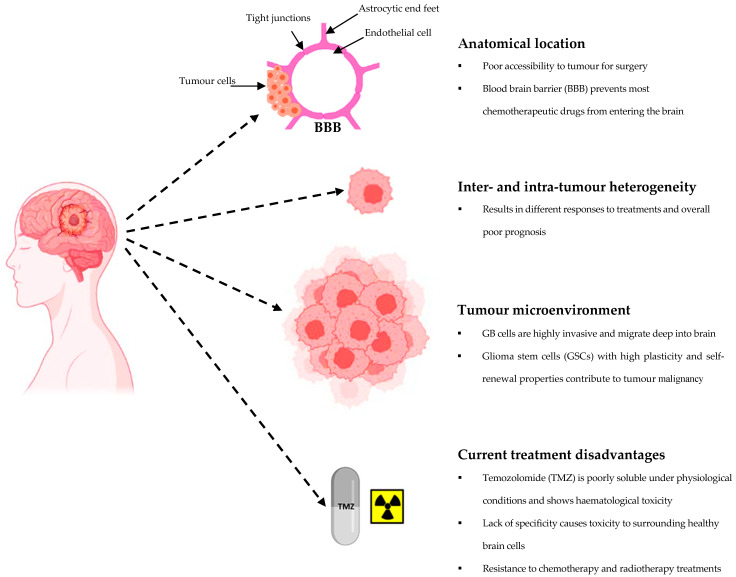
**The characteristics and therapeutic challenges associated with glioblastoma** (Figure adapted with permission from Bastiancich, Da Silva, and Estève 2021, Frontiers in Oncology, image created using BioRender.com).

**Figure 2 cancers-15-03388-f002:**
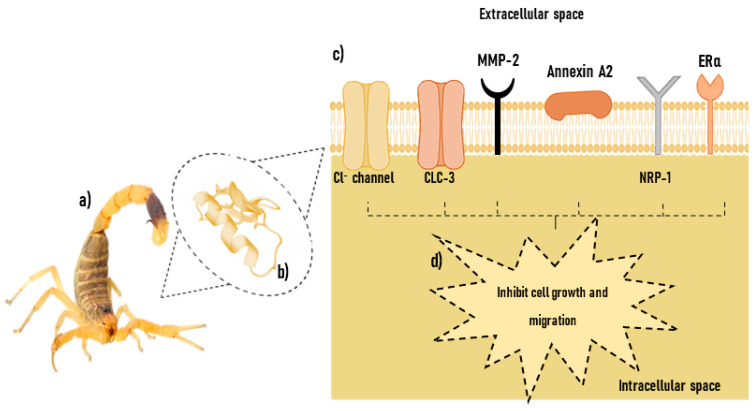
**Proposed molecular targets for CTX.** CTX is isolated from the venom of the deathstalker scorpion (*Leiurus quinquestriatus*) (**a**) and is composed of a 36-amino acid peptide stabilized by four disulfide bonds (**b**). CTX has been shown to block Cl- channels and bind to overexpressed cell surface receptors found in various tumors such as: ClC-3, MMP-2, Annexin A2, NRP-1, and ERα (**c**), all these receptors interact with CTX and ultimately contribute to an overall inhibition/suppression of cellular growth and migration (**d**) (Image created with BioRender.com).

**Figure 3 cancers-15-03388-f003:**
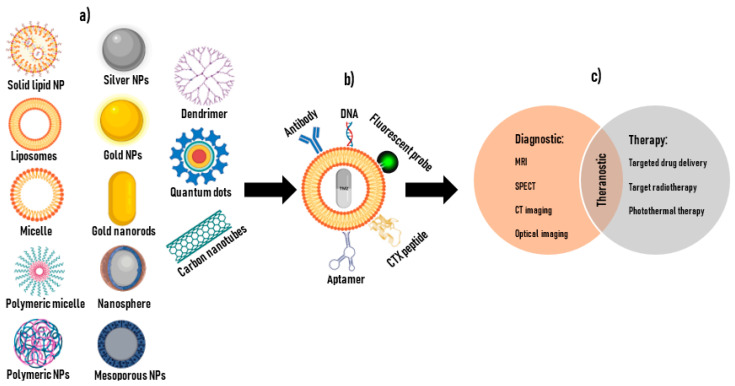
**Summary of CTX-NPs for diagnostic and therapeutic potential.** Different classes of NPs are synthesized (**a**) and designed to be target specific for cancer cells through surface functionalization with CTX peptide and other targeting molecules (**b**). CTX-NPs have applications in diagnostic and therapeutic fields, where the two overlap NPS are considered as having theranostic applications (**c**). The image was created using BioRender.com.

**Figure 4 cancers-15-03388-f004:**
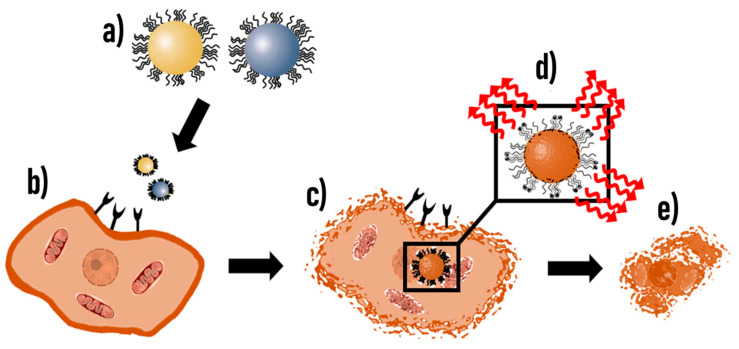
**Principle of localized nanoparticle-mediated hyperthermia in cancer cells**. Metallic NPs (e.g., AuNPs, PtNPs, or bimetallic NPs) or magnetic NPs (e.g., IONPs/SIONPs) (**a**) are designed to target specific cancer cells that overexpress specific cell surface receptors and allow intracellular uptake of NPs through receptor-mediated endocytosis (**b**). Cells are exposed to an external heating source (e.g., NIR light, AMF, and RF-fields) (**c**) which induces local heating (41–47 °C) (**d**) and results in thermal destruction of cells through mitochondrial swelling, protein denaturation, alteration in signal transduction, cell rupturing and induction of necrosis/apoptosis (**e**).

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
