# Peer review of "Diagnostic and Therapeutic Approaches for Glioblastoma and Neuroblastoma Cancers Using Chlorotoxin Nanoparticles"

_cancers, 2023, doi:10.3390/cancers15133388_

Round 1
Reviewer 1 Report
The review “Diagnostic and Therapeutic Approaches for Glioblastoma and Neuroblastoma Cancers using Chlorotoxin Nanoparticles” describes the use of chlorotoxin nanoparticles having a tropism to neuroblastoma and glioblastoma through interaction with MMP-2. Overall, the review deeply explores the therapeutic potential of these nanoparticles against brain cancer, giving a novel insight on nanomedicine-based therapies. I do recommend this review for publication after minor revision.
- Authors could implement figures. In particular, they do not show results of the enormous number of cited articles, among which they could include the principal obtained outcomes.
- Authors could briefly implement a comparison with other nanomaterials under investigation for brain cancer, such as liposomal formulations (DOI 10.3390/pharmaceutics13030378; 10.3390/ijms23063217).
- Hyperthermia section could be furtherly discussed, particularly focusing on comparisons with other nanostructures that are currently under investigation, such as MXenes or Graphene quantum dots (DOI 10.1186/s12645-023-00168-9).
Author Response
Authors could implement figures. In particular, they do not show results of the enormous number of cited articles, among which they could include the principal obtained outcomes.
Thank you for this comment. In order to address some of the comments by Reviewer 1, we have created another image, “Figure 3. Summary of CTX-NPs for diagnostic and therapeutic potential”. We have also included summaries of the principal outcomes obtained, in relevant sections of the manuscript.
Authors could briefly implement a comparison with other nanomaterials under investigation for brain cancer, such as liposomal formulations (DOI 10.3390/pharmaceutics13030378; 10.3390/ijms23063217).
Thank you for this comment. The comparison with other nanomaterials as contained in the article suggested by this reviewer, has been cited in relevant sections of the attached annotated manuscript, and marked with red fonts.
Hyperthermia section could be furtherly discussed, particularly focusing on comparisons with other nanostructures that are currently under investigation, such as MXenes or Graphene quantum dots (DOI 10.1186/s12645-023-00168-9).
Thank you for this comment. We agree with your suggestion and have included a brief discussion on MXenes or Graphene quantum dots (see fonts in red)
Reviewer 2 Report
In this manuscript, the authors systematically reviewed and discussed the characteristics of GB and NB cancers, and the potential of CTX and its functionalized NPs for diagnostic, therapeutic and theranostic purposes. This review paper may attract materials scientists, pharmaceutist and clinical doctors. Before acceptance, the following issues should be considered.
(1) More figures should be incorporated into the manuscript. For example, to help readers better understand the CTX-NPs-mediated diagnosis and therapy, schematic illustrations and/or experimental date (figures) cited from references are necessarily to be put in the right place in text.
(2) Biomimetic delivery system and CTX-loaded biomimetic nanoparticles (or living cells) should be discussed in the section of “future directions”. As biomimetic delivery system (e.g., CTX-loaded nanoparticles delivered by living cells) may successfully penetrate across the BBB.
(3) Although IONPs have been licensed for use in clinical applications by the FDA, these nanoparticles also exhibit toxicity in vivo. The toxicity of IONPs after injection should be discussed.
(4) Minor error: Page 29, the last paragraph, “Noble MNPs” may be “Noble metal”.
Author Response
More figures should be incorporated into the manuscript. For example, to help readers better understand the CTX-NPs-mediated diagnosis and therapy, schematic illustrations and/or experimental date (figures) cited from references are necessarily to be put in the right place in text.
Thank you for this comment. We have included another image to address this concern; please see Figure 3.
Biomimetic delivery system and CTX-loaded biomimetic nanoparticles (or living cells) should be discussed in the section of “future directions”. As biomimetic delivery system (e.g., CTX-loaded nanoparticles delivered by living cells) may successfully penetrate across the BBB.
Thank you for this comment. We have included a brief discussion of biomimetic NPs in two sections; check relevant sections with red font in the manuscript.
Although IONPs have been licensed for use in clinical applications by the FDA, these nanoparticles also exhibit toxicity in vivo. The toxicity of IONPs after injection should be discussed.
Thank you for this comment. We have included a short discussion on the toxicity of IONPs as suggested above; see same in relevant sections as red fonts.
Minor error: Page 29, the last paragraph, “Noble MNPs” may be “Noble metal”.
Thank you for this comment. We have corrected it.
Reviewer 3 Report
The authors systematically described the diagnostic and therapeutic approaches of CTX in GBM and NB. However, the novelty and innovative potential of your manuscript compared to the published literature should be described in more detail in the abstract and discussion section, and several issues should be further addressed:
1. The author should first introduce the clinical application and underlying mechnisms of CTX in CNS tumors in the introduction section, and the standard treatments for GB and NB (section 2 and 3) can be combined and described briefly. Also, please compare the advantages between CTX and other nanoparticles-mediated CNS tumor therapy.
2. The author should provide a figure that shows the underlying molecular mechanisms of CTX binding to these transmembrane proteins.
3. How about the structural basis for CTX binding to these proteins.
4. The in vivo toxicity and pharmacokinetics of CTX should be descried in one section.
Author Response
The author should first introduce the clinical application and underlying mechanisms of CTX in CNS tumors in the introduction section, and the standard treatments for GB and NB (section 2 and 3) can be combined and described briefly. Also, please compare the advantages between CTX and other nanoparticles-mediated CNS tumor therapy.
Thank you, the intention of the paper is to provide an update on CTX-NPs for GB and NB. Many recent reviews have provided a review on many different NPs for CTX; here our main focus was to highlight all the research that exists in the literature on CTX as a targeting molecule for NPs, specifically involving GB and NBs
The author should provide a figure that shows the underlying molecular mechanisms of CTX binding to these transmembrane proteins.
Thank you, we provided an illustration of the targeting mechanisms involved (see Fig 2)
How about the structural basis for CTX binding to these proteins?
Thank you for this comment. Fig 2 shows the proposed molecular targets for CTX, an illustration of the possible targeting mechanisms involved in CTX binding; not much has been reported in the literature with regards CTX binding mechanisms, and only a few articles have explained the potential structural basis for CTX binding to these proteins. With further research on CTX for targeting purposes, we hope that future publications will elaborate on the mechanisms.
The in vivo toxicity and pharmacokinetics of CTX should be descried in one section.
Thank you for this comment. The in vivo toxicity and pharmacokinetics of CTX has been discussed in different parts of the manuscript. Please check Sub-section “5.1. Molecular targets of CTX”; Section in 8: “Therapeutic and targeting applications of CTX-NPs for GB tumours” as well as Table 2.
The authors think that there would be no need to have a separate section on “in vivo toxicity and pharmacokinetics” as this would amount to repetition and will interfere with the flow of thought in the Sections and Sub-sections as conceived by the authors when planning the manuscript.
Round 2
Reviewer 3 Report
The revised review manuscript is acceptable now.